# Advances in Mathematical Modeling of Gas-Phase Olefin Polymerization

**Mohd Farid Atan [1,2]** , **Mohd Azlan Hussain [1],\*, Mohammad Reza Abbasi [1]** ,
**Mohammad Jakir Hossain Khan [1] and Muhamad Fazly Abdul Patah [1]**

[1]  Department of Chemical Engineering, Faculty of Engineering, University of Malaya, 50603 Kuala Lumpur, Malaysia; amfarid@unimas.my (M.F.A.); m_abbasi@ifac-mail.org (M.R.A.); jakirkhanbd@gmail.com (M.J.H.K.); fazly.abdulpatah@um.edu.my (M.F.A.P.)

[2]  Department of Chemical Engineering and Energy Sustainability, Faculty of Engineering, Universiti Malaysia Sarawak, 94300 Kota Samarahan, Malaysia

\*  Correspondence: mohd_azlan@um.edu.my; Tel.: +60-379-675-214

**Abstract:** Mathematical modeling of olefin polymerization processes has advanced significantly, driven by factors such as the need for higher-quality end products and more environmentally-friendly processes. The modeling studies have had a wide scope, from reactant and catalyst characterization and polymer synthesis to model validation with plant data. This article reviews mathematical models developed for olefin polymerization processes. Coordination and free-radical mechanisms occurring in different types of reactors, such as fluidized bed reactor (FBR), horizontal-stirred-bed reactor (HSBR), vertical-stirred-bed reactor (VSBR), and tubular reactor are reviewed. A guideline for the development of mathematical models of gas-phase olefin polymerization processes is presented.

**Keywords:** modeling; olefin; gas phase; kinetics

## 1. Introduction

Polyolefin or polyalkene, which is one of the most popular thermoplastic polymers, are formed from the monomer of the alkene group, which possesses a double bond. The most common monomers used to produce this polyolefin are ethylene and propylene. The polyolefin formed through the polymerization of ethylene is either in the form of polyethylene (PE), high-density polyethylene (HDPE), or low-density polyethylene (LDPE), and another type of polyolefin can be produced through a similar procedure from the monomer propylene in the form of polypropylene (PP). Meanwhile, copolymerization of these two monomers produces a polyolefin, which is in the form of ethylene-propylene diene monomer (EPDM). These polyolefins are in high demand and widely used in several sectors, such as automobiles, packaging, construction, and textiles. As reported by Szabó (2015) [1], the annual consumption of polyolefin per capita in the European continent is expected to increase significantly from 88 kg per person in 2015 to 120 kg per person in 2030 with an increment of 36 percent. Figure 1 summarizes the total consumption per capita in three parts of the European continent for the years 2015, 2020, and 2030.

As shown in Figure 1, Western European countries such as France, Italy, and the United Kingdom are considered to be the biggest consumers of this polyolefin compared with their neighbors located in central and Eastern Europe. This is due to the localization of several polyolefin industries, such as SO.T.AC SRL and Montello SPA in Italy, Borgeois in France, Vital Parts Ltd. in the United Kingdom, and Warm-On Gmbh, and Co.Kg in Germany. In regard to this high demand for polyolefins, the performance of the current polymerization process needs to be reviewed and improved, starting from the selection of monomers and catalyst up to the production of end products.

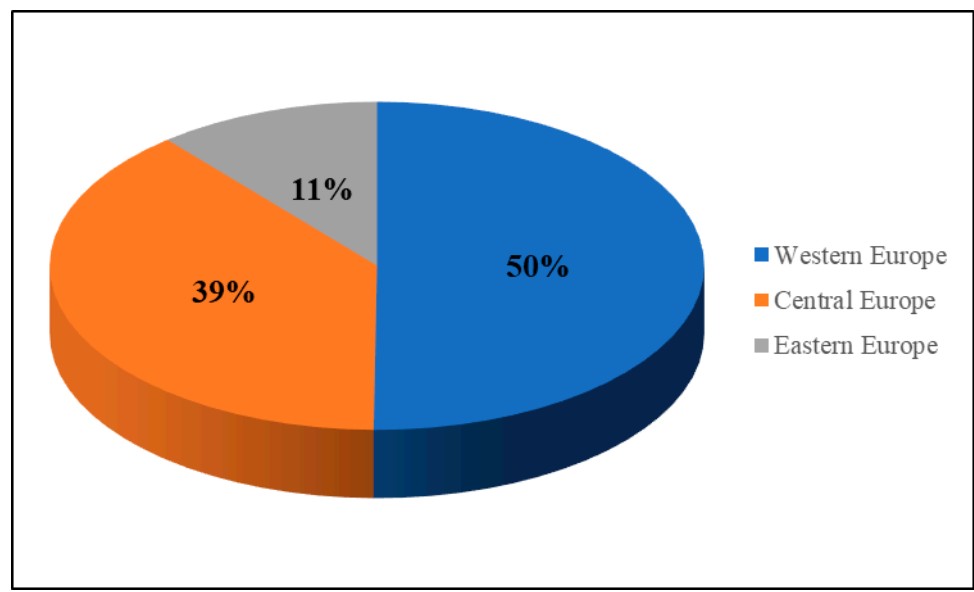

**Figure 1.** Total consumption per capita in European continent [1].

Much effort has been made by previous researchers to improve this olefin polymerization in the gas phase by reviewing different aspects, such as the thermodynamic properties, the operational conditions, the chemical processing, the reaction mechanism, the catalyst used in the polymerization reaction, and the properties of the end products. Table 1 summarizes the reviews that have been carried out previously on different aspects of gas-phase olefin polymerization.

**Table 1.** Past reviews analysis.

| No. | Review Area | Review Aspect | Ref. |
|---|---|---|---|
| 1. | Thermodynamic Properties | Different methods to determine enthalpy and entropy | [2] |
| 2. | Process Design | Design criteria, process condition, protection of instruments against overpressure, instruments for heating up and cooling down, and different types of stirrers for polymerization reactors such as autoclave reactor, high-pressure autoclave reactor, tubular reactor, fluidized bed reactor to improve the process efficiency. | [3–6] |
| 3. | Process Routes | The implementation of the solvent polymerization process, solvent polymerization without deashing, the bulk polymerization process without solvent, the vapor phase polymerization process without deashing and atactic polymer, Unipol I and II, Innovene G, Spherilene S & C, and Borstars in producing the polyolefin. | [5,7] |
| 4. | Olefin Synthesis | The implementation of free-radical methodology, carbine and nitrene methodology, and transition metal C–H bond activation methodology to synthesize the polyolefin. | [8–11] |
| 5. | Catalyst | The utilization of a metallocene catalyst system, Ziegler–Natta catalyst system, Fujita group Invented (FI) catalyst system, and oxide-supported surface organometallic complexes in olefin polymerization synthesis. | [6,12–31] |
| 6. | Process Modeling | The implementation of mathematical models, namely macroscale modeling, mesoscale modeling, microscale modeling, single particle modeling, computational fluids dynamic modeling, microelements modeling, 2D finite element modeling, single pore modeling, and parti-level fragmentation modeling to determine the properties of the polyolefin, and the mass and heat transfer phenomena during the polymerization process. | [8,11,12,22,31–34] |
| 7. | Quality Control | Different types of analysis such as nuclear magnetic resonance (NMR), temperature rising elution fractionation (TREF), gel permeation chromatography (GPC), rheological characterization (zero shear viscosity, zero shear viscosity, shear thinning behavior, dynamic modulus, loss angle, Van-Gurp-Palmen plot, Cole-Cole plot, activation energy, thermorheological complexity, strain-hardening effect, relaxation time, damping function, nonlinear dynamical oscillatory shear, and long-chain branching index), dynamic mechanical analysis, differential scanning calorimeter, neutron scattering, and molecular topology fractionation. | [12,13,22,35] |

**Table 1.** *Cont.*

| No. | Review Area | Review Aspect | Ref. |
|-----|-------------|---------------|------|
| 8. | Polyolefin Demand | The analysis of global production and consumption of polypropylene from 1985 until 2000 in the textile industry. | [36] |
| 9. | Physical and Chemical Properties of the Polyolefin | The influence of process conditions on the thermal properties, specific heat capacity, melting point, relative thermal conductivity, density, thermal diffusivity, crystallinity, amorphous phase properties, coefficient of linear thermal expansion, electrical properties, foam structure, shear, and rheological properties. | [37–39] |

From the past reviews summarized in Table 1, none of the above review studies reviewed and discussed kinetic modeling together with mass and energy balance modeling in the gas phase. The review article published by Abbasi et al. (2018) [34] only focused on a fluidized bed reactor and did not cover other types of reactors such as tubular reactor or stirred bed reactor. In addition, none of the above review studies proposed simple or proper guidelines to implement and simulate the mathematical model for this olefin polymerization process. Thus, the objective of this review is to review and discuss the past and current development of the mathematical modeling, together with the reaction mechanism of the olefin polymerization in the gas phase. This review will be concluded by proposing general guidelines in implementing and simulating the mathematical model for this olefin polymerization in the gas phase.

## 2. Mathematical Model Development for Olefin Polymerization

Theoretically, the olefin polymerization reaction occurs when the monomers, which possess reactive functional groups with double or triple bonds, are reacted in the presence of a catalyst under certain conditions of pressure and temperature. The core product formed from this reaction is called a polymer, which possesses a certain of chain length. This produced polymer can be further classified into three main chemical structures, namely, linear chain polymer, branched chain polymer, and network chain polymer. The first is also known as a thermoplastic polymer, which is formed from a monomer that possesses repeat units held by strong covalent bonds. The second is formed by a monomer, which contains the molecules in the form of a linear backbone with branches emanating randomly. Lastly, the third is also known as a gel polymer, which is formed by the extension of a branched chain polymer, which is put under reaction with high conversion [40].

Moreover, with regard to the reactor used for this polymerization process, there are two types of reactors that can be used for this olefin polymerization process, namely homogeneous and heterogeneous reactors. In the former type, the olefin polymerization occurs only in one phase and the reaction can be carried out either in a continuous stirred-tank reactor, in a loop reactor, in a hollow shaft reactor, or in a batch reactor. For the latter type, the reaction occurs in several phases such as emulsion, bubble, cloud, and solid. In this type of reaction, the analysis needs to be in each of the phases to ensure that the predicted data are coherent with the real data plant and it is highly recommended for the gas-phase olefin polymerization process. In addition, this reaction can take place either in a fluidized bed reactor or in a tubular reactor.

Furthermore, in terms of the reaction mechanism, the olefin polymerization is classified under the chain-growth polymerization where the growth of the polyolefin chain "n" occurs when the monomer is reacted with the end of the reactive functional group of the growing polyolefin under certain conditions of temperature (T) and pressure (P). It can be illustrated by using the following reaction mechanism:

$$\text{Polyolefin(n)} + \text{Monomer} \underset{\text{Catalyst}}{\overset{(T,P)}{\rightarrow}} \text{Polyolefin(n}+1)$$

Then, this chain-growth mechanism can be further sub-classified into two different categories, namely coordination and free-radical mechanisms. The former mechanism requires the use of a

coordination catalyst such as Phillips catalyst, Ziegler–Natta catalyst, and metallocene-based catalyst during the activation of active sites. Basically, this active site is made of the metal atom (Me) and the ligands (Li), which are covalently bonded. The growth of the polyolefin chain length occurs when this active site reacts with the monomer under certain conditions of temperature and pressure [41,42]. The growth mechanism of this polyolefin can be illustrated by the following reaction mechanism, where M is the monomer:

$$\text{Me} - \text{Li} \xrightarrow[\text{M}]{(T,P)} \text{Me} - \text{M} - \text{Li} \xrightarrow[\text{M}]{(T,P)} \ldots \xrightarrow[\text{M}]{(T,P)} \text{Me} - (\text{M})_n - \text{Li}$$

In contrast, the latter type of mechanism requires the presence of initiators (I) such as oxygen and organic peroxides that can be easily decomposed to form radicals. Later, these radicals will react with the monomer to grow the chain length under certain conditions of temperature and pressure [10,43]. The following free-radical mechanism is used to illustrate the growth of this olefin polymerization process.

$$\text{I} \rightarrow 2\text{R}^\bullet \xrightarrow[\text{M}]{(T,P)} \text{M} - \text{R}^\bullet \xrightarrow[\text{M}]{(T,P)} \ldots \xrightarrow[\text{M}]{(T,P)} (\text{M})_n - \text{R}^\bullet$$

The mathematical model is considered essential in predicting the output of the chemical process, in particular, the olefin polymerization process. It is written in the form of a series of algebraic and differential equations and it comprises three major elements, namely hydrodynamic, kinetic, and transport phenomena. The first phenomenon describes the process outputs such as production rate, temperature, and dynamic poly-dispersity. It allows for observing any changes or transition during the polymerization process. Then, the second phenomenon evaluates the rate or the speed of the reaction. It allows for studying the influence of input parameters or variables such as feed flow rate, type of catalyst, catalyst flow rate, pressure, inlet temperature in increasing the rate of the reaction. The last phenomenon describes the transport of momentum, energy, and chemical species via several media such as liquid, gas, and solid. The first type of transport is called fluid dynamics. The second and third types of transport are called heat and mass transfer, respectively [44].

However, for the olefin polymerization process, the modeling framework is relatively complex due to the high nonlinearity of its process dynamics caused by several factors. The first is due to the complexity of the reaction mechanisms. Theoretically, the olefin polymerization is made up of several stages, such as activation, initiation, propagation, chain transfer reaction, and deactivation. The second is caused by the complexity of the heat transfer phenomena, especially in the gas phase where it theoretically comprises several phases such as cloud, emulsion, bubble, and solid [45]. The third is caused by the physical properties of the flow behavior in both phases; solid and gas. Lastly, the fourth factor is caused by the influence of the reactor used and its operational conditions, such as the volumetric flow rate, the pressure, the reactor temperature, and the size of the catalyst particles on the final properties of the polyolefin. In addition, these final properties are measured by determining the molecular weight distribution, copolymer distribution, sequence length distribution, long chain branching distribution, short chain branching distribution, stereoregularity, and morphology properties, for instance, particle size distribution, pore size distribution, bulk density, and melt index (MI), which are among the most highlighted physicochemical properties, which have been measured frequently. To predict these physicochemical properties by using the mathematical model, there are two available methods, namely population balance modeling and method of moments that can be employed due to their ability in solving a wide range of a complex dynamic polymerization processes in the liquid and gas phases, particularly in characterizing the population of the particles as well as the growth of the polyolefin chain length [46,47].

## 2.1. Advances in the Polymerization of Olefins in the Gas Phase

Polymerization of olefins in the gas phase is considered as one of the major routes in producing polyolefins such as polypropylene, polyethylene, low-density polyethylene, ethylene-propylene copolymer, and ethylene-1-butene copolymer. A lot of effort has been made previously to model and simulate this polymerization process. Table 2 summarizes the mathematical models that have been implemented previously.

**Table 2.** Overview of implemented mathematical models.

| Model Type/Polyolefin | Process Condition | Model Assumption (s) | Ref. |
|---|---|---|---|
| Two-Phase Model/Ethylene-propylene copolymer | 1. Z–N catalyst<br>2. Fluidized Bed Reactor (FBR)<br>3. Coordination<br>4. T = [320, 500] K | 1. Emulsion does not at minimum fluidization<br>2. Well-mixed condition<br>3. The reaction occurs in the emulsion and bubble phases<br>4. Bubble and particle size are constant<br>5. Heat and mass transfer resistance is neglected<br>6. Solid elutriation is considered | [48] |
| Two-phase Model/Ethylene-1-butene copolymer | 1. Z–N catalyst<br>2. FBR<br>3. Coordination<br>4. T = [310, 317] K<br>5. P = [1.99, 2] MPa | 1. Emulsion does not at minimum fluidization<br>2. Well-mixed condition<br>3. The reaction occurs in the emulsion and bubble phases<br>4. The bubble is in a spherical form with a constant diameter<br>5. Plug flow condition with constant velocity<br>6. Heat and mass transfer resistance is neglected<br>7. Gradient temperature and concentration are neglected<br>8. Particle size distribution is uniform<br>9. Solid elutriation is considered | [49] |
| Two-Phase Model/Polypropylene | 1. Z–N catalyst<br>2. FBR<br>3. Coordination<br>4. T = [343, 353] K<br>5. P = [2, 3] MPa | 1. Eulerian–Eulerian approach<br>2. Immediate consumption of the propylene after injection the catalyst<br>3. The existence of the interaction between mass, momentum, and energy in emulsion and bubble phases<br>4. No lifts effects and virtual mass | [50] |
| Two-Phase Model/Ethylene-propylene copolymer | 1. Z–N catalyst<br>2. FBR<br>3. Coordination<br>4. T = 353.15 K<br>5. P = 2.5 MPa | 1. Emulsion does not at minimum fluidization<br>2. Well-mixed condition<br>3. The reaction occurs in the emulsion and bubble phases<br>4. Bubbles are in a spherical form with constant diameters<br>5. Plug flow condition with constant velocity<br>6. Heat and mass transfer resistance is neglected<br>7. Gradient temperature and concentration are neglected<br>8. Particle size distribution is uniform<br>9. Solid elutriation is neglected | [51] |
| Single-phase model/Polyethylene | 1. Organic peroxides and oxygen<br>2. Tubular reactor<br>3. Free radical<br>4. T = [403, 574] K<br>5. P = [152, 304] MPa | 1. Formation of a single supercritical phase<br>2. Plug flow condition<br>3. Quasi-steady-state assumption<br>4. The models depend on the ratios of kinetic rate constants<br>5. Fouling resistances at each tubular zone are uniform<br>6. No heat transfer model<br>7. The efficiencies of all initiators are similar | [52–54] |

**Table 2.** *Cont.*

| Model Type/Polyolefin | Process Condition | Model Assumption (s) | Ref. |
|---|---|---|---|
| Two-Phase Model/Polypropylene | 1. Z–N catalyst<br>2. FBR<br>3. Coordination<br>4. T = [342, 354] K<br>5. P = [2.2, 2.5] MPa | 1. Emulsion does not at minimum fluidization<br>2. Well-mixed condition<br>3. The reaction occurs in the emulsion and bubble phases<br>4. Bubbles are in a spherical form with constant diameters<br>5. Plug flow condition with constant velocity<br>6. Heat and mass transfer resistance is neglected<br>7. Gradient temperature and concentration are neglected<br>8. Particle size distribution is uniform<br>9. Solid elutriation is neglected | [55–59] |
| Single-phase Model/Low-density of Polyethylene | 1. Organic peroxides and oxygen<br>2. Tubular reactor<br>3. Free radical<br>4. T = [400, 600] K<br>5. P = [200, 300] MPa | 1. The water temperature in the cooling jacket is constant<br>2. The grade transition is influenced by the small fraction of the reaction time and is not influenced by the wall heat capacity<br>3. Steady-state assumption | [60] |
| Single-phase Model/Low-density of Polyethylene | 1. Organic peroxides and oxygen<br>2. Tubular reactor<br>3. Free radical<br>4. T = [400, 600] K<br>5. P = [200, 300] MPa | 1. Plug flow condition and single supercritical phase<br>2. Molecular weight distribution is in the form of a log-normal distribution shape<br>3. Jacket temperature and pressure in each section of the tubular reactor are constant<br>4. The process is isothermal and occurs below the gel point | [61] |
| Single-phase model/Polypropylene | 1. Z–N catalyst<br>2. FBR<br>3. Coordination<br>4. T = 353.15 K<br>5. P = 2.5 MPa | 1. The emulsion is not set at the minimum fluidization<br>2. Heat and mass transfer resistance is neglected<br>3. Dynamic monomer internal energy is neglected<br>4. Pseudo-homogeneous single phase | [62] |
| Two-Phase Model/Ethylene-1-butene copolymer | 1. Z–N catalyst<br>2. FBR<br>3. Coordination<br>4. T = [345, 374] K | 1. The reactor comprises four continuous stirred-tank reactors (emulsion phase) and four plug flow reactors (bubble phase)<br>2. Particles are in a spherical form with a constant dimension<br>3. The density function of the particles in the outlet stream and in the bed are similar | [63] |
| Three-Phase Model/Ethylene-1-butene copolymer | 1. Z–N catalyst<br>2. FBR<br>3. Coordination<br>4. T = 300 K | 1. The reaction occurs in the emulsion and solid phases<br>2. The emulsion is set at the minimum fluidization and the excess gas to maintain this condition was considered as the bubble phase<br>3. No gradient of temperature and concentration<br>4. The existence of resistance to mass transfer between the emulsion and solid phase<br>5. Rigid and porous catalyst represent the dynamic reaction<br>6. The mass transfer of emulsion molecules occurs at the surface of the solid catalyst particles | [45,64] |
| Single-phase Model/Low-density of Polyethylene | 1. Organic peroxides and oxygen<br>2. Tubular reactor<br>3. Free radical<br>4. T = [323, 604] K<br>5. P = [130, 300] MPa | 1. Plug flow condition and the supercritical reaction mixture<br>2. Existence of changes in the physical and transport properties with the axial distance<br>3. The jacket temperature at each of the reaction zones is constant<br>4. No pressure pulse<br>5. The mixture of organic peroxide and transfer agents is considered as one fictitious species<br>6. The polymer is well separated from other output components of the reactor | [65–67] |

**Table 2.** *Cont.*

| Model Type/Polyolefin | Process Condition | Model Assumption (s) | Ref. |
|---|---|---|---|
| Single-phase Model/Ethylene-1-butene Copolymer | 1. Z–N catalyst<br>2. FBR<br>3. Coordination<br>4. T = 345.15 K | 1. One serial of the continuous-stirred-tank reactor (CSTR) model<br>2. The particles are in a spherical form with constant dimensions<br>3. The attrition term in the population balance equation is constant | [68] |
| Single-phase Model/Low-density of Polyethylene | 1. Organic peroxides and oxygen<br>2. Tubular reactor<br>3. Free radical<br>4. T = [423, 574] K<br>5. P = [101, 355] MPa | 1. The flow regime is situated in the turbulent regime<br>2. At the point of injection, the pressure of the lateral feed streams and the main reaction mixture are similar<br>3. No formation of diradical and chain transfer to telogen | [69] |
| Two-Phase Model/ Ethylene-1-butene copolymer | 1. Chromium oxide catalyst<br>2. FBR<br>3. Coordination<br>4. T = 375 K | 1. Plug flow condition and quasi-steady-state assumption<br>2. Emulsion phase is homogeneous<br>3. Crystallinity and swelling are constant<br>4. The adsorbed gas phase in solid and emulsion phase is in equilibrium<br>5. The reactor bed porosity and bed porosity are identical at the minimum fluidization<br>6. Porosity condition for solid discharge is uniform<br>7. No heat loss through the fluidized bed wall<br>8. The recycle gas temperatures at the heat exchanger entrance and exit, at the reactor exit, and at the compressor exit are equal<br>9. Complete fluid back-mixing condition is considered at each of the small sub-sections in the heat exchanger<br>10. The fluid in each of the sub-sections of the heat exchanger is constant | [70] |
| Single-phase Model/High-density of Polyethylene | 1. Organic peroxides and oxygen<br>2. Extruder<br>3. Free radical<br>4. T = 443.15 K<br>5. P = 0.01 MPa | 1. Plug flow condition and the isothermal condition<br>2. One site kinetic reaction<br>3. Termination by combination is the only alternative to terminate the polymerization<br>4. Quasi-state approximation for radicals<br>5. $\gamma$ radicals are formed from the decomposition of organic peroxides via the first-order reaction | [71] |
| Single-phase Model/Low-density of Polyethylene | 1. Organic peroxides and oxygen<br>2. Tubular reactor<br>3. Free radical<br>4. T = [325, 625] K<br>5. P = [150, 250] MPa | 1. No axial mixing and temperature and concentration gradients in both reactor and jackets<br>2. The mixture is homogeneous and acts as a supercritical fluid<br>3. The temperature at each of the jacket zones is uniform | [72] |
| Two-Phase Model/ Ethylene-1-butene copolymer | 1. Z–N catalyst<br>2. FBR<br>3. Coordination<br>4. T = [310, 317] K<br>5. P = [1.99, 2] MPa | 1. No radial concentration and temperature gradients and no solid entrainment<br>2. The injection of the catalyst into the reactor as prepolymer and the mean particle size is uniform<br>3. The downward direction is considered for the overall flow direction of the polymer | [73] |
| Two-Phase Model/Polyethylene | 1. Z–N catalyst<br>2. FBR<br>3. Coordination<br>4. T = [298, 701] K<br>5. P = 0.1 MPa | 1. Simple and dynamic two-phase model and generalized bubbling–turbulent model<br>2. No variation of bubble diameter and temperature<br>3. No radial concentration gradient | [74] |

**Table 2.** *Cont.*

| Model Type/Polyolefin | Process Condition | Model Assumption (s) | Ref. |
|---|---|---|---|
| Single-phase Model/Polypropylene | 1. Z–N catalyst<br>2. FBR<br>3. Coordination<br>4. T = 343.15 K<br>5. 197. | 1. The reaction occurs only in the emulsion phase<br>2. The emulsion phase is at the minimum fluidization<br>3. Pseudo-state assumption<br>4. The catalyst is injected uniformly at each of the emulsion cells and the collection of the products is done at the bottom part of the cells<br>5. The bubble phase is formed by the excess of the fluidization gas<br>6. The number of bubble cells over the number of emulsion cells is an integer<br>7. No mass and heat transfer resistance<br>8. Solid elutriation is recycled back into the emulsion cells | [75] |
| Single-phase Model/Polypropylene | 1. Z–N catalyst<br>2. Stirred-bed reactor<br>3. Coordination<br>4. T = [330, 351] K<br>5. P = [1.9, 2.3] MPa | 1. Four CSTRs in series approximation<br>2. All the kinetic sites produce their molecular weight distribution<br>3. The catalyst possesses a single site scheme to predict the number average molecular weight in the first step and multiple site schemes in the second step | [76] |
| Two-Phase Model/ Ethylene-1-butene copolymer | 1. Z–N catalyst<br>2. FBR<br>3. Coordination<br>4. T = 345.15 K<br>5. P = 2 MPa | 1. Pseudo-homogeneous state<br>2. The emulsion phase does not remain at the minimum fluidization<br>3. No mass and heat transfer resistance between solid and emulsion gas<br>4. No mass and energy transfer resistance between emulsion and bubble phase<br>5. The reaction is at an isothermal condition<br>6. No radial temperature gradient<br>7. No solid entrainment<br>8. The catalyst is injected continuously<br>9. Mean size of the particle is uniform | [77] |
| Single-phase Model/Polypropylene | 1. Organic peroxides and oxygen<br>2. Single screw extruder<br>3. Free radical<br>4. T = 480.15 K | 1. The isothermal condition<br>2. One active kinetic site<br>3. Plug flow condition | [78] |
| Single-phase Model/High-density Polyethylene | 1. Organic peroxides and oxygen<br>2. Extruder<br>3. Free radical<br>4. T = 443.15 K<br>5. P = 0.01 MPa | 1. Only molecules with a vinyl group is present in the untreated polymer<br>2. The kinetic single site<br>3. Termination only by combination | [79] |
| Single-phase Model/Ethylene-1-butene Copolymer | 1. Z–N catalyst<br>2. FBR<br>3. Coordination<br>4. T = 360 K<br>5. P = 2.1 MPa | 1. Single-phase CSTR approximation<br>2. Perfect mixing in the beds<br>3. No radial temperature and concentration gradient<br>4. No heat and mass transfer between the solid and gas phase<br>5. Two-site kinetics scheme | [80] |
| Single-phase Model/Low-density of Polyethylene | 1. Organic peroxides and oxygen<br>2. Tubular reactor<br>3. Free radical<br>4. T = 358.15 K<br>5. P = 220 MPa | 1. Quasi-stationary state approximation<br>2. Equipment holdups are considered uniform<br>3. Plug flow condition | [81] |

**Table 2.** *Cont.*

| Model Type/Polyolefin | Process Condition | Model Assumption (s) | Ref. |
|---|---|---|---|
| Single-phase Model/Polypropylene | 1. Metallocene based catalyst<br>2. FBR<br>3. Coordination<br>4. T = 345.15 K<br>5. P = 2 MPa | 1. The reactor was divided into three main sections called annulus, draft tube and cone<br>2. No resistance mass and heat transfer between solid and gas phase<br>3. No energy generated from absorption and desorption of propylene<br>4. No gradient of velocities for solid and gas<br>5. The annulus is set at minimum fluidization<br>6. No heat transfer issued from the annulus to the draft tube<br>7. Wall and reactor temperature are equal | [82] |
| Single-phase Model/High-density Polyethylene | 1. Organic peroxides and oxygen<br>2. Extruder<br>3. Free radical<br>4. T = 443.15 K<br>5. P = 0.01 MPa | 1. The isothermal condition<br>2. Quasi-steady-state approximation<br>3. The single site kinetic scheme<br>4. No diffusion phenomena | [83] |
| Single-phase Model/Low-density Polyethylene | 1. Organic peroxides and oxygen<br>2. Tubular reactor<br>3. Free radical<br>4. T = 349.15 K<br>5. P = 228 MPa | 1. Quasi-steady-state approximation<br>2. Feedstock flow rates are constant | [84] |
| Single-phase Model/Low-density Polyethylene | 1. Organic peroxides and oxygen<br>2. Tubular reactor<br>3. Free radical<br>4. T = [323, 604] K<br>5. P = [130, 300] MPa | 1. Decomposition of oxygen into the radicals formed from the initiation is only allowed for the grouping of the activation energy with pre-exponential factor<br>2. The molecular properties are not considered in the modification of the initial parameter<br>3. Jacket temperature and pressure are different at each of the jacket zones<br>4. Plug flow and supercritical reaction mixture condition<br>5. Radial variation for the physical and transport properties<br>6. No pressure pulse | [85,86] |
| Single-phase Model/Low-density Polyethylene and Ethylene-vinyl | 1. Organic peroxides and oxygen<br>2. Tubular reactor<br>3. Free radical<br>4. T = [423, 574] K<br>5. P = [98, 196] MPa | 1. The kinetic mechanism is assumed valid for both type of polymer<br>2. No long-chain branched polymer | [87] |
| Single-phase Model/Polyethylene | 1. Z–N catalyst<br>2. FBR<br>3. Coordination<br>4. T = 300 K | 1. Well-mixed and quasi-steady-state approximation<br>2. The reaction occurs only in the dense phase<br>3. The removal flow rate is manipulated to ensure the consistency of the bed height<br>4. The catalyst was injected continuously<br>5. No influence of inert gas, co-monomer, and hydrogen | [88,89] |
| Two-Phase Model/Ethylene-1-butene copolymer | 1. Z–N catalyst<br>2. FBR<br>3. Coordination<br>4. T = 353.15 K | 1. The movement of the emulsion phase follows the plug flow regime<br>2. Particles dimensions vary<br>3. Two-site or multiple kinetic schemes<br>4. No radial gradient of concentration and temperature<br>5. No heat and mass transfer resistance between the solid and emulsion phases<br>6. No solid elutriations | [90–92] |

**Table 2.** *Cont.*

| Model Type/Polyolefin | Process Condition | Model Assumption (s) | Ref. |
|---|---|---|---|
| Single-phase Model/Ethylene-1-butene Copolymer | 1. Z–N catalyst 2. FBR 3. Coordination 4. T = 353.15 K | 1. Bubble-growth model 2. The emulsion phase is well back-mixed 3. The bubble phase comprises N well-mixed compartments in series 4. No heat and mass transfer resistance between the solid and emulsion phase 5. No reaction occurs in the bubble phase 6. No monomer mass transfer from bubble to emulsion phase 7. The emulsion phase is set at the minimum fluidization 8. Particle size varies | [93] |
| Single-phase Model/Low-density Polyethylene | 1. Organic peroxides and oxygen 2. Tubular reactor 3. Free radical 4. T = [423, 604] K 5. P = [122, 355] MPa | 1. No mass accumulation in each volume segments 2. No heat transfer due to initiation, termination and transfer reactions 3. No presence of gel effect | [94] |
| Single-phase Model/Polyethylene | 1. Z–N catalyst 2. FBR 3. Coordination 4. T = 353.15 K | 1. Well-mixed condition 2. Particle size distribution does not influence the production rate and it is discontinuous 3. The particles are unequally distributed throughout the beds 4. The agglomeration rate is influenced by the operating condition and a function of colliding particle size 5. Particles are in the form of a spherical shape with constant density and no limitation in term of inter- or intraparticle heat and mass transfer | [95] |
| Single-phase Model/Polyolefin | 1. Z–N catalyst 2. FBR 3. Coordination 4. T = 360 K | 1. The mixture is nonideal 2. No external films resistances 3. CSTR approximation 4. The deactivation has an influence on the rate constant 5. No elutriation and no particle breakage 6. The steady-state assumption | [96] |
| Single-phase Model/Low-density Polyethylene | 1. Organic peroxides and oxygen 2. Tubular reactor 3. Free radical 4. T = [323, 599] K 5. P = [182, 284] MPa | 1. Plug flow approximation 2. The generation of high-temperature peroxide 3. The utilization of water or steam as heating fluid in jackets 4. Jacket temperature and pressure are not constant 5. The presence of the friction and pressure pulse which cause the pressure drop 6. The reactivities of the telogen and monomeric radicals are equal 7. Quasi-steady-state approximation | [97] |
| Single-phase Model/Polyolefin | 1. Z–N catalyst 2. FBR, horizontal-stirred-bed reactor (HSBR), vertical-stirred-bed reactor (VSBR) 3. Coordination 4. T = 343.15 K 5. P = 2.5 MPa | 1. Total activation of catalyst since t = 0 2. The rate of initiation and propagation are similar, and higher than the rate of chain transfer 3. The only transformation is from site 1 to 2 4. Quasi-steady-state assumption 5. The occupied site is dominant | [98] |
| Single-phase Model/Polyethylene | 1. Z–N catalyst 2. FBR 3. Coordination 4. T = 273.15 K 5. P = 2.07 MPa | 1. The reactor comprises of bubble and emulsion and the reaction occurs only in the emulsion phase 2. The emulsion phase is set at the minimum fluidization 3. Bubble dimension is constant 4. The emulsion phase is back-mixed 5. No radial concentration and temperature gradients 6. No heat and mass transfer resistance between solid and gas in the emulsion phase 7. No variation in terms of the size of the particles 8. No agglomeration and elutriation of the particles 9. The steady-state approximation | [99] |

**Table 2.** *Cont.*

| Model Type/Polyolefin | Process Condition | Model Assumption (s) | Ref. |
|---|---|---|---|
| Single-phase Model/Ethylene-1-butene Copolymer | 1. Z–N catalyst<br>2. FBR<br>3. Coordination<br>4. T = 353 K<br>5. P = 3.55 MPa | 1. The size of the formed transition metal crystallites are equal<br>2. Catalyst particle is in a spherical form with a constant dimension<br>3. The multigrain solid core model<br>4. The polymer molecular weight is only influenced by the chain transfer reaction to hydrogen under isothermal conditions<br>5. Implementation of first-order deactivation kinetics for the site deactivation reaction<br>6. No intraparticle mass transfer resistance for monomers | [100] |
| Single-phase Model/Low-density Polyethylene | 1. Organic peroxides and oxygen<br>2. Tubular reactor<br>3. Free radical<br>4. T = [300, 617] K<br>5. P = 294 MPa | 1. Plug flow conditions and quasi-steady-state approximation<br>2. No variation in velocity, temperature, pressure, physical properties, and initiator efficiency<br>3. No axial mixing<br>4. No influence of viscosity on the rate constant<br>5. No heat of reaction issued from initiation, termination, and chain transfer reaction | [101] |
| Two-Phase Model/Ethylene-1-butene Copolymer | 1. Z–N catalyst<br>2. FBR<br>3. Coordination<br>4. T = 273.15 K<br>5. P = 2.07 MPa | 1. The well-mixed condition<br>2. Amorphous and gas phases are at equilibrium<br>3. No plasticizing effect of dissolved monomer<br>4. Terminal monomer or chain do not have any influence on the rate of deactivation<br>5. No radial or vertical temperature gradient<br>6. The molecular weight of ethylene and 1-butene are equal | [102] |
| Single-phase Model/Polypropylene | 1. Z–N catalyst<br>2. FBR<br>3. Coordination<br>4. T = [345, 347] K<br>5. P = [2.03, 3.55] MPa | 1. The reactor comprises of slide-free gas phase and the ideal gas law<br>2. Perfect back-mixing of gas and solid<br>3. Continuous injecting of the catalyst<br>4. The absence of net accumulation of the monomer<br>5. No amount of gas in the solid<br>6. The reactor is adiabatic | [103] |
| Single-phase Model/Low-density Polyethylene | 1. Organic peroxides and oxygen<br>2. Tubular reactor<br>3. Free radical<br>4. T = [333, 403] K<br>5. P = [193, 253] MPa | 1. Plug flow condition<br>2. Supercritical single phase is formed by the reaction mixture<br>3. Free radicals are not in steady state<br>4. The gradient of physical properties assumed to be in the axial direction | [104] |
| Single-phase Model/Polyethylene and Polypropylene | 1. Z–N catalyst<br>2. FBR<br>3. Coordination<br>4. T = 300 K | 1. Emulsion or dense phase was perfectly back-mixed<br>2. The bubbles have a constant spherical dimension<br>3. Quasi-steady-state approximation<br>4. The mass and heat transfer rate between the bubble and emulsion phase is constant<br>5. No mass and heat transfer resistance between the solid and emulsion phase | [105] |

As mentioned in Table 2, the olefin polymerization occurs via two types of mechanism, namely coordination and free-radical mechanisms. The polyolefins formed via the former mechanism in several types of reactors, namely fluidized bed reactor, vertical-stirred reactor, and horizontal-stirred-bed reactor are polypropylene, polyethylene, ethylene-propylene copolymer, and ethylene-1-butene copolymer. Meanwhile, the polyolefins formed via the latter mechanism in several types of reactors such as a tubular reactor, extruder, and autoclave reactor are low-density polyethylene, high-density polyethylene, and polypropylene. Moreover, for this type of olefin polymerization process, the reactions can be considered to occur in several phases, such as emulsion, bubble, and cloud, which differs from the olefin polymerization process in the liquid phase where the reaction can only be considered to occur in a single phase, which is the liquid or slurry phase. Furthermore, reactions that occur via a coordination mechanism requires organometallic-type catalysts such as Ziegler–Natta

catalyst, metallocene-based catalyst, or chromium oxide-based catalyst to create and activate the active sites where the olefin reaction occurs. Meanwhile, for the reaction to occur via the free-radical mechanism, it requires the presence of organic peroxide and oxygen as an initiator to create the radicals, which play a role to initiate the growth of the polymer chain.

The details of the reaction mechanisms for the coordination mechanism by using the organometallic catalyst system are tabulated in Table 3.

**Table 3.** Mechanism of the reaction for the coordination mechanism by using the organometallic catalyst.

| Reaction Mechanism | Ref. |
|---|---|
| Activation of Active Sites<br>$P^*(j) + cCata \xrightarrow{k_{act}(j)} P(0,j)$ | [45,48–51,55–59,62,64,70,75,76,80,92,93,95,98,102] |
| Spontaneous Site Activation<br>$P^* \xrightarrow{k_{actS}(j)} P(0,j)$ | [70] |
| Site Activation by Hydrogen<br>$P^*(j) + H_2 \xrightarrow{k_{actSH_2}(j)} P(0,j)$ | [70] |
| Initiation of Active Sites<br>$P(0,j) + M \xrightarrow{k_{in}(j)} P(1,j)$ | [45,48,49,51,55–59,62–64,68,70,73,76,77,80,82,92,93,95,98,100,102] |
| Propagation<br>$P(n,j) + M \xrightarrow{k_{prop}(j)} P(n+1,j)$ | [45,48–51,55–59,62–64,68,70,73,75–77,80,82,88–93,95,96,98–100,102,103,105] |
| Site Transformation<br>$P(n,j) \xrightarrow{k_{Transf}(j \to k)} P(n,k)$ | [75,98] |
| Chain Transfer to Monomer<br>$P(n,j) + M \xrightarrow{k_{trM}(j)} P(1,j) + P_d(n,j)$ | [45,48–51,55–59,62–64,68,70,73,75–77,80,88,89,91,92,95,102] |
| Chain Transfer to Hydrogen<br>$P(n,j) + H_2 \xrightarrow{k_{fH_2}(j)} P_H(0,j) + P_d(n,j)$<br>$P_H(0,j) + M \xrightarrow{k_{H_2}(j)} P(1,j)$<br>$P_H(0,j) + cCata \xrightarrow{k_{H_2C}(j)} P(1,j)$ | [45,48,49,51,55–59,62–64,68,70,73,75–77,80,88,89,91,92,95,98,100,102] |
| Chain Transfer to Co-Catalyst<br>$P(n,j) + cCata \xrightarrow{k_{trCo}(j)} P(1,j) + P_d(n,j)$ | [48–51,55–59,62,63,68,73,75–77,88,89,92,102] |
| Spontaneous Transfer<br>$P(n,j) \xrightarrow{k_{trs}(j)} P_H(0,j) + P_d(n,j)$ | [48–51,55–59,62,63,68,70,73,75–77,80,88,89,91,92,95,102] |
| Deactivation Reaction<br>$P(n,j) \xrightarrow{k_{deac}(j)} P_{deac}(0,j) + P_d(n,j)$<br>$P(0,j) \xrightarrow{k_{deac}(j)} P_{deac}(0,j)$<br>$P_H(0,j) \xrightarrow{k_{deac}(j)} P_{deac}(0,j)$ | [48–51,55–59,62,70,73,75–77,80,82,93,95,96,98,100,102] |
| Site Deactivation by Hydrogen<br>$P(n,j) + H_2 \xrightarrow{k_{deacH}(j)} P_{deac}(0,j) + P_d(n,j)$<br>$P(0,j) + H_2 \xrightarrow{k_{deacH}(j)} P_{deac}(0,j)$ | [70] |
| Site Deactivation by Oxygen<br>$P(n,j) + O_2 \xrightarrow{k_{deacO}(j)} P_{deac}(0,j) + P_d(n,j)$<br>$P(0,j) + O_2 \xrightarrow{k_{deacO}(j)} P_{deac}(0,j)$ | [70] |
| Oxygen Elimination by Alkyl Aluminum<br>$AL + O_2 \xrightarrow{k_{eO}(j)} SP$ | [70] |
| Reaction with Poisons<br>$P(n,j) + Po \xrightarrow{k_{dP}(j)} P_{dPo}(0,j) + P_d(n,j)$<br>$P_H(0,j) + Po \xrightarrow{k_{dP}(j)} P_{dPo}(0,j)$<br>$P(0,j) + Po \xrightarrow{k_{dP}(j)} P_{dPo}(0,j)$ | [48–51,55–59,62,95,96,102] |

As mentioned in Table 3, the number of stages in the reaction mechanism for this olefin polymerization process in the gas phase via the coordination mechanism by using the organometallic catalyst, namely Ziegler–Natta catalyst, metallocene-based catalyst, or chromium oxide-based catalyst implemented previously differ from one study to another study. Most of the studies considered that the polymerization reaction commences by activating the active site. At this level, the potential active sites are activated by co-catalyst to create a vacant site for the insertion of the monomer during the initiation. In addition, Salau et al. (2008) [70] proposed two additional stages, namely, spontaneous

site activation and site activation by hydrogen, which leads toward the same purpose of activating the potential active sites. After creating these vacant sites, most of the studies implemented the initiation stage, where the insertion of monomer in the vacant sites commences. At this stage, the chain length of the polyolefins starts to expand. Later, the growth of this polyolefin chain length continues during the propagation stage. After that, the growing polyolefins undergo several types of chain transfer reaction such as spontaneous transfer, chain transfer to monomer, chain transfer to hydrogen, and chain transfer to co-catalyst with the aim of controlling the molecular weight and the chain length of the polyolefin. In addition, Harshe et al. (2004) [75] and Zacca et al. (1996) [98] proposed a supplementary stage called site transformation reaction, which plays the same role as the chain transfer reactions. However, the vacant site for the insertion of the molecules is altered from the active site j to active site k located in the catalyst. Finally, to terminate the growing of the polyolefin chain length with the main aim to form the dead polyolefin, the active site or the catalyst is deactivated spontaneously or by using hydrogen, oxygen, co-catalyst, and impurities. During this catalyst deactivation, the catalytic activity and selectivity continue to decrease [27].

The mechanisms of the reaction for the olefin polymerization via free radical mechanism by using the organic peroxide and oxygen as the initiator are tabulated in Table 4.

**Table 4.** Mechanism of the reaction for the free radical mechanism by using the organic peroxide and oxygen.

| Reaction Mechanism | Ref. |
|---|---|
| Initiator Decomposition/Peroxide Initiation <br> $\text{In} \xrightarrow{k_{idec}} 2R^{\bullet}$ | [52–54,60,61,65–67,69,72,78,81,84–87,94,97,101] |
| Peroxide Initiation at High Temperature <br> $O_2 + R^{\bullet}(n) \xrightarrow{f_0 k_{idhT}} PO_2(n)$ | [86,97] |
| Generation of Peroxide at High Temperature <br> $PO_2(n) \xrightarrow{f_{P0} k_{idP0}} R^{\bullet}(n)$ | [86,97] |
| Oxygen Initiation <br> $O_2 + M \xrightarrow{k_{in}} 2R^{\bullet}$ | [65–67,72,81,84–86,97,104] |
| Thermal Initiation <br> $3M \xrightarrow{k_{ith}} R^{\bullet}(1) + R^{\bullet}(2)$ | [65–67,69,85,86,97] |
| Generation of Inert <br> $O_2 + R^{\bullet}(n) \xrightarrow{f_0 k_{ine}} Y$ | [65–67,85] |
| Initiation (Extruder) <br> $\text{In} \xrightarrow{k_{ine}} \gamma R^{\bullet} \text{ with } \gamma = 4$ | [71,79,83] |
| Chain Initiation <br> $R^{\bullet} + M \xrightarrow{k_{in}} P(1)$ | [52–54,60,61,94,101] |
| Hydrogen Abstraction (Extruder) <br> $R^{\bullet} + P(n) \xrightarrow{k_{aH}} P^{\bullet}(m) + RH$ | [83] |
| Hydrogen Abstraction without a Vinyl Group (Extruder) <br> $R^{\bullet} + P(n,j) \xrightarrow{k_{aH_2}} R^{\bullet}(n,j)$ | [71,79] |
| Hydrogen Abstraction with a Vinyl Group (Extruder) <br> $R^{\bullet} + P(n,j) \xrightarrow{k_{aH_2V}} R^{\bullet}(n,j-1)$ | [71,79] |
| Propagation <br> $R^{\bullet}(n) + M \xrightarrow{k_{prop}} R^{\bullet}(n+1)$ | [52–54,60,61,65–67,69,72,81,84–87,94,97,101,104] |
| Double Bond Propagation <br> $P(m) + R^{\bullet}(n) \xrightarrow{k_{dbprop}} R^{\bullet}(n+m)$ | [86,97] |
| Double Bond Propagation (extruder) <br> $P(n,j) + R^{\bullet}(m,k) \xrightarrow{k_{dbprop}} R^{\bullet}(n+m,j+k-1)$ | [71] |
| Chain Transfer to Monomer <br> $R^{\bullet}(n) + M \xrightarrow{k_{trM}} P(m) + R^{\bullet}(1)$ | [52–54,60,61,65–67,69,86,87,94,97,101] |
| Chain Transfer to Polymer <br> $R^{\bullet}(m) + P(n) \xrightarrow{k_{trPo}} P(m) + R^{\bullet}(n)$ | [52–54,60,65–67,69,72,78,85–87,94,97,101,104] |

**Table 4.** *Cont.*

| Reaction Mechanism | Ref. |
| --- | --- |
| Chain Transfer to Polymer (Extruder)<br>$P(m) + P^{\bullet}(n) \xrightarrow{k_{trPo}} P^{\bullet}(m) + P(n)$ | [83] |
| Chain Transfer to Polymer without a Vinyl Group (extruder)<br>$R^{\bullet}(m,k) + P(n,j) \xrightarrow{k_{trP}} P(m,k) + R^{\bullet}(n,j)$ | [71,79] |
| Chain Transfer to Polymer without a Vinyl Group (extruder)<br>$R^{\bullet}(m,k) + P(n,j) \xrightarrow{k_{trPV}} P(m,k) + R^{\bullet}(n,j-1)$ | [71,79] |
| Chain Transfer to Chain Transfer Agent/Solvent<br>$R^{\bullet}(n) + CTA \xrightarrow{k_{CTA}} P(n) + R^{\bullet}$ | [52–54,60,61,65–67,72,85–87,94,97,101,104] |
| Incorporation of Chain Transfer Agent<br>$R^{\bullet}(n) + CTA \xrightarrow{k_{iCTA}} P(n+1)$ | [52–54] |
| Termination by Combination<br>$R^{\bullet}(n) + R^{\bullet}(m) \xrightarrow{k_{tcom}} P(n+m)$ | [52–54,60,61,65–67,69,72,81,84,86,87,94,97,101,104] |
| Termination by Combination (extruder)<br>$R^{\bullet}(n,j) + R^{\bullet}(m,k) \xrightarrow{k_{tcom}} P(n+m,j+k)$ or<br>$P^{\bullet}(m) + P^{\bullet}(n) \xrightarrow{k_{tcom}} P(n+m)$ | [71,79,83] |
| Termination with Initiation Radical (extruder)<br>$R^{\bullet}(n,j) + R^{\bullet} \xrightarrow{k_{trad}} P(n,j)$ | [71] |
| Termination by Disproportionation<br>$R^{\bullet}(n) + R^{\bullet}(m) \xrightarrow{k_{tdisp}} P(n) + P(m)$ | [52–54,60,78,94,101] |
| Thermal Degradation<br>$R^{\bullet}(n) \xrightarrow{k_{thd}} P(n) + R^{\bullet}$ | [65–67,72,78,81,84–87,97,104] |
| Intramolecular Chain Transfer/Backbitting<br>$R^{\bullet}(n) \xrightarrow{k_{int}} R^{\bullet}(n)$ | [52–54,60,66,67,69,72,85,94,97,101] |
| β-Scission<br>$R^{\bullet}(n) + P(m) \xrightarrow{k_{\beta s}} P(n) + R^{\bullet}(r) + R^{\bullet}(m-r)$ | [53,54,60,78,94] |
| β-Scission (Extruder)<br>$P^{\bullet}(m) \xrightarrow{k_{\beta s}} P(n) + P^{\bullet}(m-n)$ | [83] |
| β-Scission for Sec-radicals<br>$R^{\bullet}(n) \xrightarrow{k_{\beta s2}} P(n) + R^{\bullet}$ | [52,66,67,69,72,85,97,101] |
| β-Scission for Tert-radicals<br>$R^{\bullet}(n) \xrightarrow{k_{\beta s3}} P(n) + R^{\bullet}$ | [52,66,67,72,85,97,101] |
| Retardation by the Impurities<br>$R^{\bullet}(n) + Po \xrightarrow{k_{rlm}} P(n)$ | [101] |
| Decomposition of Ethylene<br>$2C_2H_4 \xrightarrow{k_{deco}} 2C + 2CH_4 + Heat$<br>$C_2H_4 \xrightarrow{k_{deco}} 2C + 2H_2 + Heat$ | [101] |

As mentioned in Table 4, the number of stages in the reaction mechanism for this olefin polymerization process in the gas phase via a free-radical mechanism by using an organic peroxide and oxygen as the initiator implemented previously differ from one study to another study. Most of the studies considered that the polymerization reaction commences by the initiator decomposition or peroxide initiation to create the vacant site for the monomer insertion by generating the radical. In addition, these radicals can also be generated by implementing other methods, such as (i) initiation and generation of peroxide at high temperature, (ii) initiation of oxygen by reacting the oxygen with the monomer, and (iii) thermal initiation by decomposing the monomer into the radical form. Then, the radicals react with the monomer to produce the growth of the polymer chain. For the olefin reaction that takes place in the extruder, the growth of the polymer chain occurs when the radicals react with polyolefin during the abstraction of the hydrogen. After that, the growth of the polymer continues during the propagation as well as during the double bond propagation. To control the molecular weight and the length of the polyolefin, several chain transfer reactions such as chain transfer to monomer, chain transfer to polymer, chain transfer to chain transfer agent (CTA), and incorporation

of CTA are implemented. Finally, to terminate the growing of the polyolefin chain length, with the purpose of forming the dead polyolefin, the radicals are deactivated by using several methods, such as termination by combination and disproportionation, termination by initiating the radicals, thermal degradation, backbiting, β-scission, and retardation by impurities.

*2.2. Overview of the Kinetic Model, and the Mass and Energy Balance for Olefin Polymerization in the Gas Phase*

Kinetic modeling is used to predict the velocity or the speed of the chemical reaction by determining the reaction rate of the olefin polymerization reaction and it can be determined theoretically by using Equation (1) [106].

$$M + H_2 \xrightarrow{k(T)} P(0,j) \Rightarrow r = k(T)[M]^a[H_2]^b \tag{1}$$

The value of k depends on the temperature and it can be determined theoretically by using Equation (2) [107].

$$k(T) = k_0 \exp\left(\frac{-E_A}{R}\left[\frac{1}{T} - \frac{1}{T_{ref}}\right]\right) \tag{2}$$

Many studies have been carried out to determine this reaction constant experimentally to ensure the calculated reaction rate is highly accurate. The values of the reaction rate constant together with the values of activation energy ($E_A$), which are required to be used for the kinetic modeling and simulation for the organometallic-type catalyst system such as Ziegler–Natta catalyst, metallocene-based catalyst, and chromium oxide-based catalyst are mentioned in the following publications [45,49,51,55–59,62,68,73–76,80,82,88,89,91,93,95,98–100,102,103,105,108–110]. For the olefin polymerization via the free-radical mechanism, using organic peroxide and oxygen as the initiator, the values of reaction rate constants, together with their corresponding activation energy, were also published [61,69,71,72,78,83,85,94,97,101,104].

Several mathematical models have been implemented to describe the dynamic behavior of this olefin polymerization process in the gas phase. By referring to Table 2, most of the mathematical models were simulated by using a fluidized bed reactor, followed by the tubular reactor, extruder, autoclave reactor, horizontal-stirred-bed reactor, and vertical-stirred-bed reactor. The fluidized bed reactor, vertical and horizontal-stirred-bed reactors were used to carry out the olefin polymerization reaction by using a Ziegler–Natta catalyst, chromium oxide-based catalyst, and a metallocene-based catalyst system. Meanwhile, the tubular reactor, the extruder, and the autoclave reactor were used to perform the olefin polymerization reaction by using the initiator, namely, organic peroxide and oxygen. To summarize, even though the reaction was carried out in the same type of reactor or by using the same type of catalyst or initiator, the proposed mathematical models slightly differed due to the different considerations in the number of stages in the reaction mechanism and the assumptions to simplify the complexity of the nonlinearity phenomena that occurred during olefin polymerization in the gas phase.

For the olefin synthesis via a coordination mechanism, the following kinetic model can be used to determine theoretically the number of moles of the potential sites and the initiation sites, the population balance for living and dead chains, the moment of the chain length distribution for the living and dead polymer, and the population balance for dead polymer [49,56,58,59,62]. For the number of moles of the potential sites, it is written in the following form:

$$\frac{dP^*(j)}{dt} = F_{in}^*(j) - k_{act}(j)P^*(j) - P^*(j)\frac{R_v}{V_p} \tag{3}$$

Then, the number of moles of the initiation sites can be determined using the following formulas:

$$\frac{dP(0,j)}{dt} = k_{act}(j)P^*(j) - P(0,j)\left\{k_{deac}(j)[M] + k_{ds}(j) + k_{dP}(j)[Po] + \frac{R_v}{V_P}\right\} \tag{4}$$

$$\frac{dP_H(0,j)}{dt} = Y(0,j)\left\{k_{fH_2}(j)[H_2] + k_{trs}(j)\right\} - N_H(0,j)\left\{k_{H_2}(j)[M] + k_{deac}(j)\right.$$
$$\left. + k_{H_2C}(j)[AL] + k_{dP}[Po] + \frac{R_v}{V_P}\right\} \tag{5}$$

Moreover, to determine the population balance for a living and dead chains polymer for chain length equal to 1 and greater than 2, the following formulas can be used:

$$\frac{dP(1,j)}{dt} = k_{act}(j)P(0,j)[M] + P_H(0,j)\left\{k_{H_2}(j)[M] + k_{H_2C}(j)[AL]\right\}$$
$$+ Y(0,j)\left\{k_{trM}(j)[M] + k_{trCo}(j)[AL]\right\} - N(1,j)\left\{k_{prop}(j)[M]\right.$$
$$\left. + k_{trM}(j)[M] + k_{fH_2}(j)[H_2] + k_{trCo}(j)[AL] + k_{trs}(j) + k_{deac}(j) + k_{dP}(j)[Po] + \frac{R_v}{V_P}\right\} \tag{6}$$

$$\frac{dP(n,j)}{dt} = k_{prop}(j)[M]P(n-1,j) - P(n,j)\left\{k_{prop}(j)[M] + k_{trM}(j)[M]\right.$$
$$\left. + k_{trCo}(j)[AL] + k_{fH_2}(j)[H_2] + k_{trs}(j) + k_{deac}(j) + k_{dP}(j)[Po] + \frac{R_v}{V_P}\right\} \tag{7}$$

$$\frac{dQ(1,j)}{dt} = P(1,j)\left\{[M]k_{trM}(j) + [H_2]k_{fH_2}(j) + [AL]k_{trCo}(j)\right.$$
$$\left. + k_{trs}(j) + k_{deac}(j) + k_{dP}(j)[Po]\right\} - \frac{R_v}{V_P}Q(n,j) \tag{8}$$

$$\frac{dQ(n,j)}{dt} = P(n,j)\left\{[M]k_{trM}(j) + [H_2]k_{fH_2}(j) + [AL]k_{trCo}(j)\right.$$
$$\left. + k_{trs}(j) + k_{deac}(j) + k_{dP}(j)[Po]\right\} - \frac{R_v}{V_P}P(n,j) \tag{9}$$

Lastly, the following equations can be used to calculate the zeroth, first and second moment of the chain length distribution for the living and dead polymer:

$$\frac{dY(0,j)}{dt} = k_{act}(j)P(0,j)[M] + P_H(0,j)\left\{k_{H_2}(j)[M] + k_{H_2C}(j)[AL]\right\}$$
$$- Y(0,j)\left\{k_{fH_2}(j)[H_2] + k_{trs}(j) + k_{deac}(j) + k_{dP}(j)[Po] + \frac{R_v}{V_P}\right\} \tag{10}$$

$$\frac{dY(1,j)}{dt} = k_{act}(j)P(0,j)[M] + P_H(0,j)\left\{k_{H_2}(j)[M] + k_{h_2C}(j)[AL]\right\}$$
$$+ Y(0,j)\left\{k_{trM}(j)[M] + k_{trCo}(j)[AL]\right\} + k_{prop}[M]Y(0,j)$$
$$- Y(1,j)\left\{k_{trM}(j)[M] + k_{fH_2}(j)[H_2] + k_{trCo}(j)[AL] + k_{trs}(j) + k_{deac}(j)\right.$$
$$\left. + k_{dP}(j)[Po] + \frac{R_v}{V_P}\right\} \tag{11}$$

$$\frac{dY(2,j)}{dt} = k_{act}(j)P(0,j)[M] + P_H(0,j)\left\{k_{H_2}(j)[M] + k_{H_2C}(j)[AL]\right\}$$
$$+ Y(0,j)\left\{k_{trM}(j)[M] + k_{trCo}(j)[AL]\right\} + k_{prop}[M]\left\{2Y(1,j) + Y(0,j)\right\}$$
$$- Y(2,j)\left\{k_{trM}(j)[M] + k_{fH_2}(j)[H_2] + k_{trCo}(j)[AL] + k_{trs}(j) + k_{deac}(j) + k_{dP}(j)[Po] + \frac{R_v}{V_P}\right\} \tag{12}$$

$$\frac{dX(n,j)}{dt} = Y(n,j)\left\{[M]k_{trM}(j) + [H_2]k_{fH_2}(j) + [AL]k_{trCo}(j)\right.$$
$$\left. + k_{trs}(j) + k_{deac}(j) + k_{dP}(j)[Po]\right\} - \frac{R_v}{V_P}X(n,j) \tag{13}$$

Furthermore, there exist several types of heat and mass transfer models implemented previously to describe the dynamic behavior of this polymerization process. For a three-phase model where the reactions were assumed to occur in emulsion, cloud, and bubble phases, the heat and mass transfer models are written in the following form [45]. The mass balance for the potential sites, active sites, and catalyst are written in the following form:

$$\frac{dP^*(j)}{dt} = \frac{Q_{cat}P_{in}^*(j)}{m_s} - \frac{Q_{vprod}P^*(j)\rho_{cat}}{m_s} - R_{iP^*} \tag{14}$$

$$\frac{dP(0,j)}{dt} = \frac{Q_{cat}P(0,j)}{m_s} - \frac{Q_{vprod}P(0,j)\rho_{cat}}{m_s} - R_{iP(0,j)} \tag{15}$$

$$\frac{d[Cat]}{dt} = \frac{Q_{cat}}{m_s} - \frac{Q_{vprod}[Cat]\rho_{cat}}{m_s} \tag{16}$$

Furthermore, for the heat and mass balances from the bubble phase to the cloud phase, it can be determined by using the following formula:

$$u_b\frac{d[R_A]_b}{dz} = -K_{bc}([R_A]_b - [R_A]_c) \text{ with } z = 0, \ [R_A]_b = [R_A]_{b0} \tag{17}$$

$$\frac{d([R_A]_b(T_b - T_{ref}))}{dz} = \frac{H_{bc}}{u_bC_{p,g}}(T_c - T_b) \tag{18}$$

Then, for the heat and mass balance from the cloud phase to the emulsion phase, the equations are written in the following form:

$$u_b\delta\left[\frac{3\left(\frac{u_{mf}}{\varepsilon_{mf}}\right)}{u_b - \frac{u_{mf}}{\varepsilon_{mf}}} + \alpha\right]\frac{d[R_A]_b}{dz} = K_{bc}([R_A]_b - [R_A]_c) - K_{ce}([R_A]_c - [R_A]_e) \tag{19}$$

$$z\frac{d([R_A]_c(T_c - T_{ref}))}{dz} = \frac{H_{ce}}{u_bC_{p,g}}(T_e - T_c) \tag{20}$$

Moreover, for the heat and mass transfer with chemical reaction from the emulsion phase to the catalyst phase, the equations are formulated in the following form:

$$\begin{aligned} A_{bed}(He)\varepsilon_{mf}\frac{d[R_A]_e}{dt} &= K_{ce}([R_A]_c - [R_A]_e)A_{bed}(He)\varepsilon_{mf} + Q_{vm}([R_A]_0 - [R_m]_e) \\ &\quad - Q_{vprod}[R_m]_e\varepsilon_{mf} + r_Am_s \end{aligned} \tag{21}$$

$$\begin{aligned} A_{bed}(He)&\left[(1 - \varepsilon_{mf})\rho_sC_{p,s} + \varepsilon_{mf}[R_A]_{mf}C_{p,g}\right]\frac{dT_e}{dt} + A_{bed}(He)(T_e - T_{ref})\varepsilon_{mf}C_{p,g}\frac{d[R_m]_e}{dt} = \\ &-Q_{vm}[R_m]_eC_{p,g}(T_e - T_f) + A_bH_{be}\int(T_b - T_e)dz + (-\Delta H_r)r_A \\ &-Q_{vprod}\varepsilon_{mf}[R_m]_eC_{p,g}(T_e - T_{fs}) - Q_{vprod}\varepsilon_{mf}[R_m]_eC_{p,g}(T_e - T_f) \\ &-\pi D(He)(1 - \delta^*)h_w(T_e - T_w) \end{aligned} \tag{22}$$

Finally, to determine the population balance for a living and dead polymer for this three-phase model, the following equations can be used:

$$\frac{dY(n,j)}{dt} = R_{Y(n,j)} - \frac{Q_{vprod}Y(n,j)\rho_{cat}}{m_s} \tag{23}$$

$$\frac{dX(n,j)}{dt} = R_{X(n,j)} - \frac{Q_{vprod}X(n,j)\rho_{cat}}{m_s} \tag{24}$$

Then, for a two-phase model where the reactions were assumed to occur in the emulsion and bubble phase, the heat and mass transfer model is written in the following form [48]. For heat and mass balance equations in the emulsion phase, the equations are written as follows:

$$\begin{aligned} \frac{d(V_e\varepsilon_e[M]_e)}{dt} &= [M]_{e,in}U_eA_e - [M]_eU_eA_e - R_{ve}\varepsilon_e[M]_e \\ &\quad + K_{be}([M]_b - [M]_e)V_e\left(\frac{\delta}{1-\delta}\right) \\ &\quad - (1 - \varepsilon_e)R_{i_e} - \frac{K_eV_e\varepsilon_eA_e[M]_e}{W_e} \end{aligned} \tag{25}$$

$$\frac{d(V_e\,\varepsilon_e\,[H_2]_e)}{dt} = [H_2]_{e,in}U_eA_e - [H_2]_eU_eA_e - R_{v_e}\varepsilon_e[H_2]_e$$
$$+K_{be}([H_2]_b - [H_2]_e)V_e\left(\frac{\delta}{1-\delta}\right) \tag{26}$$
$$-(1-\varepsilon_e)R_{i_e} - \frac{K_eV_e\,\varepsilon_e\,A_e[H_2]_e}{W_e}$$

$$\left(V_e\left(\varepsilon_e\sum_{i=1}^{m}C_{pi}[M_i]_e + (1-\varepsilon_e)\rho_{poly}C_{p,pol}\right)\right)\frac{d(T_e-T_{ref})}{dt} =$$
$$U_eA_e(T_{e,in} - T_{ref})\sum_{i=1}^{m}[M_i]_{e,in}C_{pi} - U_eA_e(T_e - T_{ref})\sum_{i=1}^{m}[M_i]_eC_{pi}$$
$$-R_{ve}(T_e - T_{ref})\left(\sum_{i=1}^{m}\varepsilon_eC_{pi}[M_i]_e + (1-\varepsilon_e)\rho_{poly}C_{p,pol}\right) + (1-\varepsilon_e)R_{pe}\Delta H_R \tag{27}$$
$$-H_{be}V_e\left(\frac{\delta}{1-\delta}\right)(T_e - T_b) - V_e\varepsilon_e(T_e - T_{ref})\sum_{i=1}^{m}C_{pi}\frac{d[M_i]_e}{dt}$$
$$-\frac{K_eA_e}{W_e(T_e-T_{ref})}\left(\sum_{i=1}^{m}\varepsilon_eC_{pi}[M_i]_e + (1-\varepsilon_e)\rho_{poly}C_{p,pol}\right)$$

Meanwhile, the equations for heat and mass transfer in the bubble phase are written as follows:

$$\frac{d(V_b\,\varepsilon_b\,[M]_b)}{dt} = [M]_{b,in}U_bA_b - [M]_bU_bA_b - R_{vb}\varepsilon_b[M]_b - K_{be}([M]_b - [M]_e)V_b$$
$$-(1-\varepsilon_b)\frac{A_b}{V_{PFR}}\int R_{i_b}dz - \frac{K_bV_b\,\varepsilon_b\,A_b[M]_b}{W_b} \tag{28}$$

$$\frac{d(V_b\,\varepsilon_b\,[H_2]_b)}{dt} = [H_2]_{b,in}U_bA_b - [H_2]_bU_bA_b - R_{vb}\varepsilon_b[H_2]_b - K_{be}([H_2]_b - [H_2]_e)V_b$$
$$-(1-\varepsilon_b)\frac{A_b}{V_{PFR}}\int R_{i_b}dz - \frac{K_bV_b\,\varepsilon_b\,A_b[H_2]_b}{W_b} \tag{29}$$

$$\left(V_b\left(\varepsilon_b\sum_{i=1}^{m}C_{pi}[M_i]_b + (1-\varepsilon_b)\rho_{poly}C_{p,pol}\right)\right)\frac{d(T_b-T_{ref})}{dt} =$$
$$U_bA_b(T_{b,in} - T_{ref})\sum_{i=1}^{m}[M_i]_{b,in}C_{pi} - U_bA_b(T_b - T_{ref})\sum_{i=1}^{m}[M_i]_bC_{pi}$$
$$-R_{vb}(T_b - T_{ref})\left(\sum_{i=1}^{m}\varepsilon_bC_{pi}[M_i]_b + (1-\varepsilon_b)\rho_{poly}C_{p,pol}\right) \tag{30}$$
$$+(1-\varepsilon_b)\frac{A_b\Delta H_R}{V_{PFR}}\int R_{pb}dz + H_{be}V_b(T_e - T_b) - V_b\varepsilon_b(T_b - T_{ref})\sum_{i=1}^{m}C_{pi}\frac{d[M_i]_b}{dt}$$
$$-\frac{K_bA_b}{W_b(T_b-T_{ref})}\left(\sum_{i=1}^{m}\varepsilon_bC_{pi}[M_i]_b + (1-\varepsilon_b)\rho_{poly}C_{p,pol}\right)$$

Furthermore, for single-phase known as a well-mixed model where the reactions were assumed to occur in the emulsion phase, the following heat and mass transfer model can be implemented [99,102]. The heat and mass balance equations are formulated as follows:

$$V\varepsilon_{mf}\frac{d[M]}{dt} = U_0A([M]_{in} - [M]) - R_v\varepsilon_{mf}[M] - (1-\varepsilon_{mf})R_i \tag{31}$$

$$V\varepsilon_{mf}\frac{d[H_2]}{dt} = U_0A([H_2]_{in} - [H_2]) - R_v\varepsilon_{mf}[H_2] - (1-\varepsilon_{mf})R_i \tag{32}$$

$$\left(\sum_{i=1}^{m}[M_i]C_{pi}V\varepsilon_{mf} + V(1-\varepsilon_{mf})\rho_{pol}C_{p,pol}\right)\frac{dT}{dt} =$$
$$U_0A\sum_{i=1}^{m}[M_i]_{in}C_{pi}(T_{in} - T_{ref}) - U_0A\sum_{i=1}^{m}[M_i]C_{pi}(T - T_{ref}) \tag{33}$$
$$-R_v\left[\sum_{i=1}^{m}[M_i]C_{pi}\varepsilon_{mf} + (1-\varepsilon_{mf})\rho_{pol}C_{p,pol}\right](T - T_{ref}) + (1-\varepsilon_{mf})\Delta H_R R_p$$

Lastly, another single-phase model is known as constant bubble size model, the following heat and mass transfer model was previously implemented [105]. For heat and mass balance in the bubble phase, the equations are written as follows:

$$[\overline{M}]_b = \frac{1}{H} \int_0^{He} [M]_b dh = [M]_e + \left([M]_{e,in} - [M]_e\right) \frac{U_b}{K_{be}He} \left(1 - e^{-\left(\frac{K_{be}He}{U_b}\right)}\right) \tag{34}$$

$$[\overline{H_2}]_b = \frac{1}{H} \int_0^{He} [H_2]_b dh = [H_2]_e + \left([H_2]_{e,in} - [H_2]_e\right) \frac{U_b}{K_{be}He} \left(1 - e^{-\left(\frac{K_{be}He}{U_b}\right)}\right) \tag{35}$$

$$\overline{T}_b = \frac{1}{He} \int_0^H T_b dh = T_e + (T_{in} - T_e) \frac{U_b \overline{C}_p}{H_{be}He} \left(1 - e^{-\left(\frac{H_{be}H}{U_b \overline{C}_p}\right)}\right) \tag{36}$$

with

$$\overline{C}_p = \sum_{i=1}^{Nm} [\overline{M}_i]_b C_{pMi} = [M]_b C_{pC_3H_6} + [H_2]_b C_{pH_2} + [N_2]C_{pN_2} \tag{37}$$

For the emulsion phase, the heat and mass transfer equations are written as follows:

$$V_e \varepsilon_{mf} \frac{d[M]_e}{dt} = U_e A_e \varepsilon_{mf} \left([M]_{e,in} - [M]_e\right) + \frac{V_e \delta K_{be}}{(1-\delta)} \left([\overline{M}]_b - [M]_e\right) - R_{ve} \varepsilon_{mf} [M]_e - (1 - \varepsilon_{mf}) R_i \tag{38}$$

$$\begin{aligned} V_e \varepsilon_{mf} \frac{d[H_2]_e}{dt} &= U_e A_e \varepsilon_{mf} \left([H_2]_{e,in} - [H_2]_e\right) + \frac{V_e \delta K_{be}}{(1-\delta)} \left([\overline{H_2}]_b - [H_2]_e\right) \\ &\quad - R_{ve} \varepsilon_{mf} [H_2]_e - (1 - \varepsilon_{mf}) R_i \end{aligned} \tag{39}$$

$$\begin{aligned} &\left(\sum_{i=1}^{m} V_e \varepsilon_{mf} [M_i]_e C_{pi} + V_e (1 - \varepsilon_{mf}) \rho_{pol} C_{p,pol}\right) \frac{dT_e}{dt} = \\ &- \sum_{i=1}^{m} V_e \varepsilon_{mf} C_{pi} \frac{d[M_i]_e}{dt} (T_e - T_{ref}) + U_e A_e \varepsilon_{mf} \sum_{i=1}^{m} [M_i]_{e,in} C_{pi} (T_{e,in} - T_{ref}) \\ &- U_e A_e \varepsilon_{mf} \sum_{i=1}^{m} [M_i]_e C_{pi} (T_e - T_{ref}) - \frac{V_e \delta H_{be}}{(1-\delta)} \left(T_e - \overline{T}_b\right) \\ &+ R_{ve} \left((1 - \varepsilon_{mf}) \rho_{pol} C_{p,pol} + \varepsilon_{mf} \sum_{i=1}^{m} [M_i]_e C_{pi}\right) (T_e - T_{ref}) + (1 - \varepsilon_{mf}) \Delta H_R R_{pe} \end{aligned} \tag{40}$$

For the olefin synthesis via a free-radical mechanism, the following equations can be used to determine mass and heat transfer models (radicals, polymer, monomer, reactor, and reactor jacket), pressure drop, kinetic model (moment of dead polymer), mass balance for long and short chain branching polymer [61,69]. For the mass balance for the radicals, the equation is written as follows:

$$\begin{aligned} \frac{d[R(n)]}{dt} &= 2fk_{idec}[In] - k_{prop}[M]([R(n)] - [R(n-1)](1 - \delta_{n,1})) \\ &- (k_{CTA}[CTA] + k_{trM}[M]) \left([R(n)] - \delta_{n,1} \sum_{i=1}^{\infty} [R(i)]\right) - k_{tcom}[R(n)] \sum_{i=1}^{\infty} [R(i)] \end{aligned} \tag{41}$$

Then, the mass balance for the polymer is formulated as follows:

$$\frac{d[P(n)]}{dt} = (k_{CTA}[CTA] + k_{trM}[M])[R(n)] + \frac{k_{tcom}}{2} \sum_{m=1}^{n-1} [R(m)][R(n-m)](1 - \delta_{n,1}) \tag{42}$$

Moreover, the mass balance for the monomer used during the synthesis is written as follows:

$$\frac{d[M]}{dt} = -(k_{prop} + k_{trM})[M] \sum_{i=1}^{\infty} [R(i)] \tag{43}$$

Meanwhile, for the heat balance for the reactor and the cooling fluid in the reactor jacket, the equations are written as follows:

$$\frac{dT}{dt} = \frac{(-\Delta H_R)LR_{pm}}{\rho u C_p} + \frac{uA_{pipe}(T - T_{cool})}{\rho u A_{sp}C_p} \tag{44}$$

$$\frac{dT_{cool}}{dt} = \pi D_{out}LU\frac{(T_{cool} - T)}{Q_m C_{p,cool}} \tag{45}$$

Furthermore, the pressure drop occurred in the reactor can be determined as follows:

$$\frac{dP}{dz} = -L\left(2f_{ric}\rho\frac{u^2}{D_{in}} + \rho u\frac{du}{dz}\right) \text{ with } f_{ric}^{-1/2} = 4\log\left(f_{ric}^{1/2}Re\right) - 0.4 \tag{46}$$

Then, to determine the zeroth, first and second moments of dead polymer, the equations are formulated as follows:

$$\frac{d(X(0)P(n))}{dz} = \frac{k_{tcom}Y^2(0)}{2} + (k_{trM}[M] + k_{\beta s2})Y(0) \tag{47}$$

$$\frac{d(X(1)P(n))}{dz} = k_{tcom}Y(0)Y(1) + (k_{trM}[M] + k_{\beta s2})Y(1) \\ + k_{trPo}\left((Y(1) + Y'(1))X(1) - X(2)\sum_{i=1}^{\infty}[R(i)]\right) \tag{48}$$

$$\frac{d(X(2)P(n))}{dz} = k_{tcom}\left(Y(0)Y(2) + Y^2(1)\right) + (k_{trM}[M] + k_{\beta s2})Y(2) \\ + k_{trPo}\left((Y(2) + Y'(2))X(1) - X(3)\sum_{i=1}^{\infty}[R(i)]\right) \tag{49}$$

In addition, for the zeroth, first and second moments of temporary dead polymer I and II, the equations are written as follows:

$$\frac{d\left(X'(0)P(n)\right)}{dz} = k_{tcom}Y(0)Y'(0) - k_{idec}Y'(0) + (k_{trM}[M] + k_{\beta s2})Y'(0) \tag{50}$$

$$\frac{d\left(X'(1)P(n)\right)}{dz} = k_{tcom}\left(Y(0)Y'(1) + Y(1)Y'(0)\right) - k_{idec}Y'(1) + (k_{trM}[M] + k_{\beta s2})Y'(1) \\ + k_{trPo}\left((Y(1) + Y'(1))X'(1) - X'(2)\sum_{i=1}^{\infty}[R(i)]\right) \tag{51}$$

$$\frac{d\left(X'(2)P(n)\right)}{dz} = k_{tcom}\left(Y(2)Y'(0) + 2Y(1)Y'(1) + Y(0)Y'(2)\right) - k_{idec}Y'(2) + (k_{trM}[M] + k_{\beta s2})Y'(2) \\ + k_{trPo}\left((Y(2) + Y'(2))X'(1) - X'(3)\sum_{i=1}^{\infty}[R(i)]\right) \tag{52}$$

$$\frac{d(X''(0)P(n))}{dz} = k_{tcom}Y''(0) - 2k_{idec}Y''(0) \tag{53}$$

$$\frac{d(X''(1)P(n))}{dz} = k_{tcom}Y'(0)Y'(1) - 2k_{idec}Y''(1) \tag{54}$$

$$\frac{d(X''(2)P(n))}{dz} = k_{tcom}\left(Y'(0)Y'(2) + Y''^2(1)\right) - 2k_{idec}Y''(2) \tag{55}$$

Finally, the mass balance for short and long chain branching polymer is defined as follow:

$$\frac{d[P_{scb}(n)]}{dz} = k_{scb}\sum_{i=1}^{\infty}[R(i)] \tag{56}$$

$$\frac{d[P_{lcb}(n)]}{dt} = k_{lcb}\left(X(1) + X'(1)\right)\sum_{i=1}^{\infty}[R(i)] \tag{57}$$

By referring to Equations (3) until 57, a method of moments and population balances, which are in form of ordinary differential equation (ODEs), were used to build and simulate the kinetics model with the aim of studying the dynamic behavior of the olefin polymerization reaction in a fluidized bed reactor, horizontal-stirred-bed reactor, vertical-stirred-bed reactor, tubular reactor, extruder, and autoclave reactor via two types of reaction mechanisms, namely, coordination and free-radical mechanisms. For the olefin polymerization via a coordination mechanism, Equations (58) to (60) are used to calculate the polydispersity index, number average molecular weight, and the weight average molecular weight, respectively.

$$\text{PDI} = \frac{M_w}{M_n} \tag{58}$$

$$M_n = \frac{\sum\limits_{j=1}^{NS} Y(1,j)X(1,j)}{\sum\limits_{j=1}^{NS} Y(0,j)X(0,j)} MW \tag{59}$$

$$M_w = \frac{\sum\limits_{j=1}^{NS} Y(2,j)X(2,j)}{\sum\limits_{j=1}^{NS} Y(1,j)X(1,j)} MW \tag{60}$$

The polymerization rate is defined as follows [59]:

$$R_p = MW[M]Y(0,j)k_{prop}(j) \tag{61}$$

The melt index or melt flow index (MFI) can be determined by using the following equation [45]:

$$\text{MFI} = 3.346 \times 10^{17} M_w^{-3.472} \tag{62}$$

For the olefin polymerization process via a free-radical mechanism, to determine the number and weight average molecular weight, the following equations are used, which are slightly different from the equation used for the coordination mechanism. At this level, the moments of dead and temporary dead polymer, as well as the moments of living polymer, are incorporated to determine these parameters [69]:

$$M_w = MW\frac{X(2) + X'(2) + X''(2) + Y(2) + Y'(2)}{X(1) + X'(1) + X''(1) + Y(1) + Y'(1)} \tag{63}$$

$$M_n = MW\frac{X(1) + X'(1) + X''(1) + Y(1) + Y'(1)}{X(0) + X'(0) + X''(0) + Y(0) + Y'(0)}$$

Lastly, the monomer conversion for this olefin polymerization via a free-radical mechanism is defined by the following equation:

$$y_{mono} = 1 - \frac{[M]u}{[M]_0} \tag{64}$$

To compute the mass and heat balance equations, which represent the olefin reaction via a coordination mechanism, the hydrodynamic correlations that are tabulated in Abbasi et al. (2016) [49] are referred. For the heat and mass balance equations for the olefin polymerization process via a free-radical mechanism, the hydrodynamic correlations that are tabulated in Khazraei and Dhib (2008) [69] are used.

### 2.3. Numerical Methods for the Simulation of the Mathematical Model

Equation (3) until 57 are either in the form of ODEs or in the form of partial differential equations (PDEs). For the ODEs, they contain only one independent variable, which is generally time (t). The PDEs contains at least two independent variables, which are generally time, height of the reactor, etc. To simulate these types of equations, several methods exist, which are summarized in Table 5.

**Table 5.** Numerical methods for the process simulation.

| Methods | Main Feature | Ref. |
|---------|--------------|------|
| Euler's Method | Ability to solve simple and linear ordinary differential equation (ODEs) with the presence of initial values | [111–113] |
| Monte Carlo Method | Ability to compute the ODEs with random values | [113,114] |
| Rosenbrock Method | Ability to solve stiff ODEs | [113,115,116] |
| Backward Euler's Method | Ability to solve stiff ODEs with larger step size | [117] |
| Finite Difference Method | Ability to solve partial differential equations (PDEs) by approximating the nonlinear system to linear system | [118] |
| Method of Lines | Ability to solve PDE by approximating the PDE system with an ODE system. In general, the spatial independent variables are substituted by algebraic approximation (as a function of time) | [119] |
| Finite Element Method | Ability to solve PDEs with the presence of boundary conditions | [120] |
| Multigrid Methods | Ability to solve high order PDEs, especially parabolic systems. | [121] |

By referring to Table 5, because of the stiffness of the ODEs used in this polymerization process, Rosenbrock and backward Euler's method seem to be the most suitable numerical methods to be implemented in simulating these ODEs [58]. Meanwhile, for the PDEs, because the independent variables are not more than two, the finite difference method or method of lines could be applied.

### 3. A General Guideline to Implement the Mathematical Model

After reviewing the mathematical model, a general guideline can be proposed to ease the implementation of the mathematical model for the olefin polymerization process in the gas phase. The following flowchart in Figure 2 can be used to illustrate the procedure.

For olefin polymerization in the gas phase, two types of major mechanism occur in the process. If the free-radical mechanism (details can be found in Table 4) occurs, the normal choice is to choose a tubular reactor, while for the coordination mechanisms (details can be found in Table 3), the fluidized bed reactors and stirred-bed reactors (horizontal and vertical) are the preferred choices. The next step is to decide on the type of model to be used based on the number of phases assumed in the reactor. For the tubular reactor, the model to be used is normally the single-phase model assuming all the reactions occurs homogeneously in the packed bed tubular reactor. For the fluidized bed reactors (FBR) and stirred-bed reactors (SBR), the model can be either single, two or three phase depending on the assumptions of the location of the sites of the reactions occurring in the process. Examples of single, two, and three phases can be seen in Table 2, with different types of process conditions and assumptions by different researchers.

The final mathematical model for the mass balance (either 1, 2, or 3 phases) will incorporate the kinetic model, transport phenomena process, and the hydrodynamics within the reaction system. The kinetic model includes the method of moments, while the effects of population balance can also be

included in the mass balance. A suitable model will also be done for the heat balance incorporating the kinetic model and transport phenomena mechanisms.

Finally, these mathematical models (both the mass and energy balance) can be validated using computational fluid dynamics (CFD) models (ANSYS 6.1, ANSYS Inc., Berkeley, CA, USA) and through experimental pilot plant data [56].

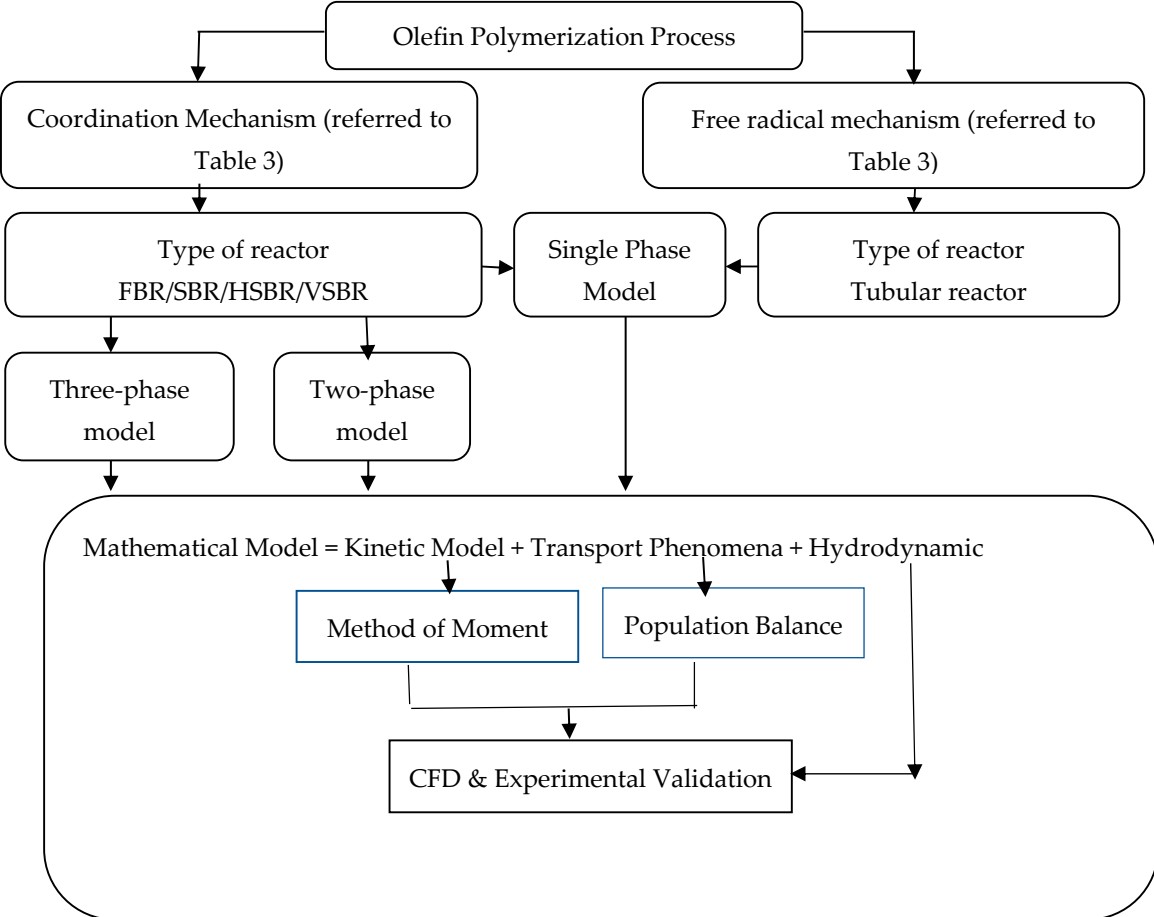

**Figure 2.** A general flowchart of the guidelines for the implementation of the mathematical model (FBR: fluidized bed reactor; SBR: stirred-bed reactors; HSBR: horizontal-stirred-bed reactor; VSBR: vertical-stirred-bed reactor).

## 4. Conclusions

Polyolefins such as polyethylene or polypropylene are widely used nowadays in producing several materials for industrial and consumer use. To ensure that the end products are safe, high quality and have an optimized production facility, the processing conditions need to be improved. Thus, a review of the previously implemented gas-phase mathematical models was carried out to scaffold a methodology to build new mathematical models or to improve the existing ones. For the olefin polymerization process in the gas phase, the tubular reactor operates at high pressures while reactors such as a fluidized bed reactor, stirred-bed reactor, vertical-stirred-bed reactor and horizontal-stirred-bed reactor work at moderate pressures. Moreover, most of the modeling in olefin polymerization via a free-radical mechanism considers a single-phase model. Meanwhile, for the olefin polymerization in the gas phase via a coordination mechanism, several models exist, such as the three-phase model, two-phase model, well-mixed (single-phase) model, and constant bubble size model (two-phase with constant bubble size). To simplify this complexity, some researchers ignore extra phases during model implementation. Many studies have been done to improve the precision

of the two-phase model as well as single-phase models by adding more details to the model, such as considering solid elutriation. In the future, studies on heat loss through fluidized bed reactors and improvement in the precision of the three-phase model can be carried out. The flowchart in Figure 2 can be used as a guideline to retrofit the available models or to develop new ones. A lot of work has been done on modeling olefin polymerization, but a lot of effort is still needed to validate the models better, which is to minimize the gap between model output and experimental or industrial data with the aim of improving the end products that are environmentally friendly and have a high quality.

**Author Contributions:** M.F.A. reviewed all the published articles related to olefin polymerization in gas phase and prepared this review manuscript. Meanwhile, M.A.H., M.R.A., M.J.H.K. and M.F.A.P. contribute in reviewing and enriching the content of this review article.

**Funding:** This research received no external funding.

**Acknowledgments:** Authors would like to acknowledge the Ministry of Education Malaysia and Universiti Malaysia Sarawak (UNIMAS) in funding this research through public academician scholarship scheme. Last but not least, the authors would like to thank the University of Malaya in providing the research facilities to execute this research.

**Conflicts of Interest:** The authors declare no conflict of interest

## Nomenclature

| | |
|---|---|
| a | Partial rate constants |
| A | Area ($m^2$) |
| $A_{tube}$ | The cross-sectional area of the cooling jacket tube ($m^2$) |
| $A_{bed}$ | The cross-sectional area of the fluidized bed ($m^2$) |
| $A_{pipe}$ | The surface area of the pipe ($m^2$) |
| $A_{sp}$ | The cross-sectional area of the pipe ($m^2$) |
| AL | Alkyl Aluminum ($mol/m^3$) |
| b | Partial rate constants |
| BP | By-product ($mol/m^3$) |
| $C_i$ | Concentration of the component i ($mol/m^3$) |
| cCata | Co – Catalyst ($mol/m^3$) |
| $C_p$ | Heat capacity (J/Kg·K) |
| $C_{p,pol}$ | Heat capacity of the polyolefin (J/Kg·K) |
| $C_{p,tube}$ | Heat capacity in the cooling jacket tube (J/Kg·K) |
| D | Diameter (m) |
| $D_{in}$ | Inlet diameter (m) |
| $D_{out}$ | Outer diameter (m) |
| $D_z$ | Dispersion coefficient ($m^2/s$) |
| $E_A$ | Activation energy (J/mol) |
| $F_m$ | Mass flow rate (kg/s) |
| $F_{in}^*(j)$ | Potential active site flow rate of a site type j injected into the reactor (mol/s) |
| $F_{m,tube}$ | Mass flow rate in the jacket cooling tube (kg/s) |
| $F_{cat}$ | The fraction of the catalyst in the polyolefin |
| $F_{cata}$ | Mass fraction of the activated catalyst |
| $F_{dcata}$ | Mass fraction of the deactivated catalyst |
| $F_{H_2}$ | Mass fraction of the hydrogen |
| $F_M$ | Mass fraction of the monomer |
| FBR | Fluidized bed reactor |
| $f_0 k_{ine}$ | Generation of the inert rate constant ($m^3 mol^{-1} s^{-1}$) |
| He | Height of the reactor (m) |
| H | Enthalpy of the reactor (J/kg) |
| $H_{be}$ | Bubble to emulsion heat transfer coefficient ($W/m^3 K$) |
| $H_{bc}$ | Bubble to cloud heat transfer coefficient ($W/m^3 K$) |
| $H_{ce}$ | Cloud to emulsion heat transfer coefficient ($W/m^3 K$) |

| | |
|---|---|
| $H_{in}$ | Enthalpy of the inlet feedstock into the reactor (J/kg) |
| $H_2$ | Hydrogen (mol/m$^3$) |
| $h_w$ | Wall heat transfer coefficient (W/m$^2$K) |
| HSBR | Horizontal-stirred bed reactor |
| In | Initiators (mol/m$^3$) |
| $k_0$ | Pre-exponential reaction rate constant (s$^{-1}$ or m$^3$mol$^{-1}$s$^{-1}$) |
| $k$ | Reaction rate constant |
| $k_{act}(j)$ | Activation rate constant for active site type j (m$^3$mol$^{-1}$s$^{-1}$) |
| $k_{actH_2}(j)$ | Catalyst activation by hydrogen rate constant for a site type j (s$^{-1}$) |
| $k_{actM}(j)$ | Catalyst activation by monomer rate constant for a site type j (s$^{-1}$) |
| $k_{actS}(j)$ | Spontaneous site activation rate constant for active site type j (s$^{-1}$) |
| $k_{actSA}(j)$ | Site activation by alkyl aluminum rate constant for active site type j (s$^{-1}$) |
| $k_{actSH_2}(j)$ | Site activation by hydrogen rate constant for a site type j (s$^{-1}$) |
| $k_{actSM}(j)$ | Site activation by monomer rate constant for a site type j (s$^{-1}$) |
| $k_{aH}$ | Hydrogen abstraction rate constant (m$^3$mol$^{-1}$s$^{-1}$) |
| $k_{aH_2}$ | Hydrogen abstraction without a vinyl group rate constant (m$^3$mol$^{-1}$s$^{-1}$) |
| $k_{aH_2V}$ | Hydrogen abstraction with a vinyl group rate constant (m$^3$mol$^{-1}$s$^{-1}$) |
| $k_{dP}(j)$ | Reaction with poisons rate constant (m$^3$mol$^{-1}$s$^{-1}$) |
| $k_{Deac}(j)$ | Deactivation rate constant (s$^{-1}$) |
| $k_{DPo}(j)$ | Deactivation by poison rate constant (s$^{-1}$) |
| $k_{deac}(j)$ | Deactivation rate constant for a site type j (s$^{-1}$) |
| $k_{deco}$ | Decomposition of ethylene rate constant (s$^{-1}$) |
| $k_{fH_2}(j)$ | Chain transfer to hydrogen rate constant for a site type j with terminal monomer M reacting with hydrogen (m$^3$mol$^{-1}$s$^{-1}$) |
| $k_{idec}$ | Initiator decomposition rate constant (s$^{-1}$) |
| $f_0k_{idhT}$ | Peroxide initiator at high-temperature rate constant (m$^3$mol$^{-1}$s$^{-1}$) |
| $f_{P0}k_{idP0}$ | Peroxide generation at high-temperature rate constant (s$^{-1}$) |
| $k_{in}$ | Initiation rate constant (m$^3$mol$^{-1}$s$^{-1}$) |
| $k_{in}(j)$ | Initiation rate constant for a site type j (m$^3$mol$^{-1}$s$^{-1}$) |
| $k_{ine}$ | Initiation rate constant in extruder (m$^3$mol$^{-1}$s$^{-1}$) |
| $k_{ith}$ | Thermal initiation rate constant (s$^{-1}$) |
| $k_{prop}$ | Propagation rate constant (m$^3$mol$^{-1}$s$^{-1}$) |
| $k_{prop}(j)$ | Propagation rate constant for a site type j (m$^3$mol$^{-1}$s$^{-1}$) |
| $k_{spont}(j)$ | Spontaneous chain transfer rate constant for a site type j (m$^3$mol$^{-1}$s$^{-1}$) |
| $k_{tcom}$ | Termination by combination rate constant (m$^3$mol$^{-1}$s$^{-1}$) |
| $k_{tdisp}$ | Termination by disproportionation rate constant (m$^3$mol$^{-1}$s$^{-1}$) |
| $k_{thd}$ | Thermal degradation rate constant (s$^{-1}$) |
| $k_{Transf}(j \rightarrow k)$ | Site transform from site j to site k rate constant (s$^{-1}$) |
| $k_{trad}$ | Termination with initiation radical rate constant (m$^3$mol$^{-1}$s$^{-1}$) |
| $k_{trCTA}$ | Chain transfer to chain transfer agent rate constant (m$^3$mol$^{-1}$s$^{-1}$) |
| $k_{trCo}$ | Chain transfer to co-catalyst rate constant (m$^3$mol$^{-1}$s$^{-1}$) |
| $k_{trCo}(j)$ | Chain transfer to co-catalyst rate constant for a site type j (m$^3$mol$^{-1}$s$^{-1}$) |
| $k_{trH_2}(j)$ | Chain transfer to hydrogen rate constants for a site type j (m$^3$mol$^{-1}$s$^{-1}$) |
| $k_{trs}(j)$ | Chain transfer to solvent rate constants for a site type j (m$^3$mol$^{-1}$s$^{-1}$) |
| $k_{trM}(j)$ | Chain transfer to monomer rate constant for a site type j (m$^3$mol$^{-1}$s$^{-1}$) |
| $k_{trM}$ | Chain transfer to monomer rate constant (m$^3$mol$^{-1}$s$^{-1}$) |
| $k_{trs}(j)$ | Spontaneous transfer rate constant (m$^3$mol$^{-1}$s$^{-1}$) |
| $k_{trP}$ | Chain transfer to the polymer without a vinyl group rate constant (m$^3$mol$^{-1}$s$^{-1}$) |
| $k_{trPV}$ | Chain transfer to a polymer with a vinyl group rate constant (m$^3$mol$^{-1}$s$^{-1}$) |
| $k_{trPo}$ | Chain transfer to polymer rate constant (m$^3$mol$^{-1}$s$^{-1}$) |
| $k_{\beta s}$ | β-scission rate constant (m$^3$mol$^{-1}$s$^{-1}$) |
| $k_{\beta s2}$ | β-scission for secondary radical rate constant (s$^{-1}$) |
| $k_{\beta s3}$ | β-scission for tertiary radical rate constant (s$^{-1}$) |

| | |
|---|---|
| $K_z$ | Heat dispersion coefficient (J/m·s·K) |
| $K_{be}$ | bubble to emulsion mass transfer coefficient (s$^{-1}$) |
| $K_{bc}$ | Bubble to cloud mass transfer coefficient (s$^{-1}$) |
| $K_{ce}$ | Cloud to emulsion mass transfer coefficient (s$^{-1}$) |
| $K_e$ | Elutriation constant in emulsion phase (kg/m$^2$s) |
| L | Length of the reactor (m) |
| M | Monomer used during the polymerization (mol/m$^3$) |
| MFI | Melt flow index or Melt index (g/min) |
| MW | Molecular weight (kg/mol) |
| $M_n$ | Number average molecular weight (kg/mol) |
| $M_w$ | Weight average molecular weight (kg/mol) |
| m | Mass inside the reactor (kg) |
| $m_{poly}$ | Mass of the polymer inside the reactor (kg) |
| Me | Metal atoms (mol) |
| $N_s$ | Number of active sites j |
| $O_2$ | Oxygen (mol/m$^3$) |
| P | Pressure (Pa) |
| Po | Poison (mol/m$^3$) |
| $P^*(j)$ | Potential active site of type j (mol) |
| $P(0,j)$ | Uninitiated site of type j produced from activation reaction (mol) |
| $P_H(0,j)$ | Uninitiated site of type j produced from chain transfer to hydrogen reaction (mol) |
| $P(1)$ | Living polymer chain with a chain length one produced by the initiation reaction (mol) |
| $P(1,j)$ | Living polymer chain of type j with a chain length one produced by the initiation reaction (mol) |
| $P(m)$ | Growth living chain with a chain length m with the terminal monomer M (mol) |
| $P(n)$ | Growth living chain with a chain length n with the terminal monomer M (mol) |
| $P(n+m)$ | Growth living chain with a chain length n + m with the terminal monomer M (mol) |
| $P(n+m,j+k)$ | Growth living chain of type j + k with a chain length n + m (mol) |
| $P(n,j)$ | Growth living polymer chain of type j with a chain length n with the terminal monomer M (mol) |
| $P(m,k)$ | Growth living polymer chain of type k with a chain length m with the terminal monomer M (mol) |
| $P(n+1)$ | Growth living chain with a chain length n + 1 with the terminal monomer M (mol) |
| $P(n+1,j)$ | Growth living polymer chain of type j with a chain length n + 1 with the terminal monomer M (mol) |
| $P_d(n)$ | Dead polymer chain (mol) |
| $P_d(n,j)$ | Dead polymer chain of type j (mol) |
| $P_{dPo}(0,j)$ | Impurity site of type j (mol) |
| $P_{deac}(j)$ | Deactivated site of type j (mol) |
| PDI | Polydispersity index |
| Po | Impurity (mol/m$^3$) |
| q | Heat transfer via the cooling jacket (J/s) |
| $Q(n,j)$ | Dead polymer with n chain length of type j (mol) |
| $Q_{cat}$ | The mass flow rate of the catalyst (kg/s) |
| $Q_{H_2}$ | The mass flow rate of the hydrogen (kg/s) |
| $Q_{in}$ | Inlet flow rate (kg/s) |
| $Q_{Mon}$ | The mass flow rate of the monomer (kg/s) |
| $Q_m$ | The mass flow rate (kg/s) |
| $Q_{out}$ | Outlet flow rate (kg/s) |
| $Q_{outF}$ | Outflow rate of the polyolefin in the slurry phase (kg/s) |
| $r_A$ | Rate expression for the active sites (mol/kg catalyst per second) |

| | |
|---|---|
| $R_A$ | Reactant used during polymerization process |
| $R^\bullet$ | Radical (mol) |
| $R^\bullet(1)$ | Living radical with the chain length 1 (mol) |
| $R^\bullet(2)$ | Living radical with the chain length 2 (mol) |
| $R^\bullet(m)$ | Living radical with m chain length (mol) |
| $R^\bullet(n)$ | Living radical with n chain length (mol) |
| $R^\bullet(n,j)$ | Living radical with n chain length of type j (mol) |
| $R^\bullet(m,k)$ | Living radical with n chain length of type k (mol) |
| $R^\bullet(n+1)$ | Living radical with n + 1 chain length (mol) |
| $R^\bullet(m-r)$ | Living radical with m – r chain length (mol) |
| $R^\bullet(m)$ | Living radical with m chain length (mol) |
| RH | Inert molecule (mol) |
| R | Gas constant (J/mol·K) |
| $R_d$ | Deactivation reaction rate (kg/s) |
| Re | Reynold Number |
| $R_{H_2}$ | Hydrogen consumption rate (kg/s) |
| $R_p$ | Polymerization rate (kg/s) |
| $R_{pm}$ | Polymerization rate (mol/L·s) |
| $R_v$ | Volumetric production rate of polymer ($m^3 s^{-1}$) |
| SP | Sub-product (mol/$m^3$) |
| T | Temperature (K) |
| $T_{tube}$ | Cooling jacket tube temperature (K) |
| $T_{ref}$ | Reference temperature (K) |
| U | Heat transfer constant or internal energy (W/$m^2$K) |
| u | Velocity (m/s) |
| V | Volume ($m^3$) |
| $V_p$ | The volume of polymer phase present in the reactor ($m^3$) |
| VSBR | Vertical-stirred bed reactor |
| $V_g$ | The volume of the gas ($m^3$) |
| $V_{tube}$ | The volume of the liquid in the cooling jacket tube ($m^3$) |
| WSBR | Well-stirred semi-batch reactor |
| W | The weight of particle solid (kg) |
| X(0) | Zeroth moment of chain length distribution of the dead polymer (mol) |
| X(1) | The first moment of chain length distribution of the dead polymer (mol) |
| X(2) | The second moment of chain length distribution of the dead polymer (mol) |
| X(0,j) | Zeroth moment of chain length distribution of the dead polymer chain (mol) |
| X(1,j) | The first moment of chain length distribution of the dead polymer chain (mol) |
| X(2,j) | The second moment of chain length distribution of the dead polymer chain (mol) |
| X(n,j) | The n moment of chain length distribution of the dead polymer chain (mol) |
| Y | Inert molecule (mol) |
| Y(0) | Zeroth moment of chain length distribution of the living polymer (mol) |
| Y(1) | The first moment of chain length distribution of the living polymer (mol) |
| Y(2) | The second moment of chain length distribution of the living polymer (mol) |
| Y(0,j) | Zeroth moment of chain length distribution of the living polymer chain (mol) |
| Y(1,j) | The first moment of chain length distribution of the living polymer chain (mol) |
| Y(2,j) | The second moment of chain length distribution of the living polymer chain (mol) |
| $y_{mono}$ | Monomer conversion |
| $Z-N$ | Ziegler-Natta |

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
