# Peer review of "Advances in Mathematical Modeling of Gas-Phase Olefin Polymerization"

_processes, doi:10.3390/pr7020067_

Reviewer 1 Report

This paper presents a review on the past and currently implemented mathematical models of the olefin polymerization process in gas phase via coordination and free radical mechanism in several types of reactors such as fluidized bed reactor, horizontal-stirred bed reactor, vertical-stirred bed reactor and tubular reactor. A guideline is also given to facilitate the development of the mathematical model of gas phase olefin polymerization.

The paper needs proof read to eliminate some typo and grammatical errors, e.g. remove “The” from “The Figure 1” in the first page, in page 2, “Located in 42 central and eastern Europe” is not a complete sentence and should be combined with the previous sentence.

Tables 3 and 4 with a lot of repeated references does not read well. Perhaps it is better to list the reactions with common references first and then mention these reactions with special references.

There is no need to put these mathematical equations in Table 5. They can just be listed in the main text.

It would be better to add some discussions on numerical methods for simulate the process.

Author Response

Point 1: The paper needs proofread to eliminate some typo and grammatical errors, e.g. remove “The” from “The Figure 1” in the first page, in page 2, “Located in 42 central and eastern Europe” is not a complete sentence and should be combined with the previous sentence.

Response 1: This paper has been proofread by Dr. McNaught, editor appointed by Cambrige Proofreading LLC (Refer to the attached document).

Point 2: Tables 3 and 4 with a lot of repeated references does not read well. Perhaps it is better to list the reactions with common references first and then mention these reactions with special references.

Response 2: None of the reaction mechanism possesses the common references after carrying out the matrix analysis of the reaction mechanism versus the references. Thus, the reaction mechanism with the highest number of references is analyzed further to regroup the references. The references which appear always in the same mechanism of the reaction is regrouped under one group. The arrangements for the coordination mechanism are as follows:

a.       [A] = [48], [49], [51], [55], [56], [57], [58], [59], [62], [103]

b.      [B] = [71], [81]

c.       [C] = [74], [78]

d.      [D] = [63], [69]

e.       [E] = [64], [65]

f.        [F] = [76], [77]

g.      [G] = [91], [100], [104], [106]

The arrangements for the free-radical mechanism are as follows:

a.       H] = [67], [68]

b.      [I] = [52], [102]

c.       [J] = [53], [54], [60], [95]

d.      [K] = [66], [87]

e.       [L] = [82], [85]

Point 3: There is no need to put these mathematical equations in Table 5. They can just be listed in the main text.

Response 3: All the equations have been listed in the text (see page 23 until 29) and Table 5 has been removed.

Point 4: It would be better to add some discussions on numerical methods to simulate the process.

Response 4: Several numerical methods such as Euler’s method, Monte Carlo method, Rosenbrock method, backward Euler’s method, finite difference method, method of lines, finite element method, and multigrid methods are listed in new Table 5 with some discussion.

Note: The details of the correction are mentioned in the attached document

Reviewer 2 Report

The work entitled "A Review of the Development in Mathematical Modeling of Gas Phase Olefin Polymerization" presents a well-justified new review on the cited topic, which for me is a little specific for a review but the high number of references are out of any doubt. After carefully reading the work, I consider it is suitable to be published as is in Processes. 

The background is well-documented as well as the novelty of this review. Although I do not support long tables in a review, I consider a well-organized work because the information is appended in tables and authors comments are briefly given below with surprising effectiveness for the readership. About the content, all issues are well described and are easy presented to the readership. In other words, I have no contributions to append in this review.

Author Response

Point: The background is well-documented as well as the novelty of this review. Although I do not support long tables in a review, I consider a well-organized work because the information as appended in tables and authors’ comments are briefly given below with the surprising effectiveness for the readership. About the content, all issues are well described and are easily presented to the readership. In other words, I have no contributions to append in this review.

Response: No amendment needs to be carried out
